# Structural basis of 3′-tRNA maturation by the human mitochondrial RNase Z complex

Genís Valentín Gesé [iD][1] & B Martin Hällberg [iD][1,2✉]

## Abstract

**Maturation of human mitochondrial tRNA is essential for cellular energy production, yet the underlying mechanisms remain only partially understood. Here, we present several cryo-EM structures of the mitochondrial RNase Z complex (ELAC2/SDR5C1/ TRMT10C) bound to different maturation states of mitochondrial tRNA[His], showing the molecular basis for tRNA-substrate selection and catalysis. Our structural insights provide a molecular rationale for the 5′-to-3′ tRNA processing order in mitochondria, the 3′-CCA antideterminant effect, and the basis for sequence-independent recognition of mitochondrial tRNA substrates. Furthermore, our study links mutations in ELAC2 to clinically relevant mitochondrial diseases, offering a deeper understanding of the molecular defects contributing to these conditions.**

**Keywords** Mitochondria; RNA Processing; RNase Z; ELAC2; Cryo-EM
**Subject Categories** RNA Biology; Structural Biology

## Introduction

All human RNA transcripts must be enzymatically processed in different ways to fulfill their biological functions. In the nucleus, RNA polymerase III transcribes tRNA precursors containing 5′- and 3′-leader sequences, which are cleaved by the RNase P complex and RNase Z, respectively, to generate tRNAs and other regulatory non-coding RNAs (Altman et al, 1993; Dieci et al, 2007; Siira et al, 2018).

In contrast to nuclear tRNAs, mitochondrial tRNAs are transcribed within two polycistronic heavy- and light-strand precursor RNA transcripts containing the coding sequences for mRNAs, rRNAs, and tRNAs. Most RNA moieties in the polycistronic transcripts are separated by tRNA sequences, which serve as landmarks to direct the endonucleolytic cleavage of the polycistronic transcripts by the mitochondrial RNase P and RNase Z complexes (tRNA-punctuation model (Ojala et al, 1981).

Nuclear and mitochondrial RNase P are evolutionarily and architecturally distinct in humans (Sridhara, 2024). Specifically, human nuclear RNase P is a large RNA–protein complex in which the RNA part catalyzes RNA processing (Bartkiewicz et al, 1989),

whereas human mitochondrial RNase P is a protein-only enzyme and acts as a complex consisting of TRMT10C/SDR5C1/PRORP (Holzmann et al, 2008). In this complex, PRORP is the nuclease subunit (Holzmann et al, 2008; Reinhard et al, 2015; Vilardo et al, 2023). TRMT10C is a purine N1-methyltransferase that acts at position 9 of mt-tRNAs using S-adenosylmethionine (SAM) as a co-substrate (Vilardo et al, 2012). SDR5C1 is a moonlighting homotetrameric short-chain dehydrogenase/reductase (Holzmann et al, 2008; Vilardo and Rossmanith, 2015).

After tRNA 5′-processing, 3′-processing by RNase Z ensues. In all domains of life, RNase Z belongs to the metallo-β-lactamase family of enzymes and comprises long and short forms. In bacteria and archaea, RNase Z is present only in the short form, which contains one fully functional metallo-β-lactamase domain and functions as a dimer. The long form, which is present along with the short form in many eukaryotes, results from gene duplication and contains two metallo-β-lactamase domains, the NTD and the CTD, where only the CTD has retained the HxHxDH, the PxKxRN (P-loop) and the AxDx motifs important for the enzymatic activity while the NTD contains an insertion known as the exosite that consists of a compact globular domain formed by an insertion in the β-lactamase domain and lies at the tip of a two-stranded stalk (Vogel et al, 2005). In humans, the short-form homolog is named ELAC1 and localizes in the cytosol and the nucleus (Brzezniak et al, 2011; Rossmanith, 2011; Takahashi et al, 2008). The human long form of RNase Z, ELAC2, has two alternative start codons to produce a nuclear and a mitochondrially targeted form (Brzezniak et al, 2011; Rossmanith, 2011).

Mutations in ELAC2 are coupled to many clinically relevant mitochondrial diseases, with patient symptoms ranging from pediatric cardiomyopathy to adult intellectual disability depending on their functional severity (Akawi et al, 2016; Cafournet et al, 2023; Paucar et al, 2018; Saoura et al, 2019). ELAC2 function is also important for nuclear processing of tRNAs and other non-coding RNAs such as sno-RNAs (Siira et al, 2018). To the best of our knowledge, no clinically relevant mutations in ELAC2 have been reported related explicitly to nuclear RNA processing, presumably because mitochondrial dysfunction dominates the clinical picture.

We have previously shown that after 5′-leader cleavage by PRORP, most human mitochondrial tRNA precursors remain bound to TRMT10C/SDR5C1, significantly enhancing the efficiency of the ensuing ELAC2 3′-processing for 17 of the 22 human mitochondrial tRNAs (Reinhard et al, 2017). Remarkably, the tRNA precursors also remain bound to TRMT10C/SDR5C1 after

[1]Department of Cell and Molecular Biology, Karolinska Institutet, Solna, Sweden. [2]Centre for Structural Systems Biology (CSSB) and Karolinska Institutet VR-RÅC, Hamburg, Germany. ✉E-mail: Martin.Hallberg@ki.se

ELAC2 processing and even during further maturation. We, therefore, proposed a model in which the TRMT10C/SDR5C1 complex acts as a tRNA maturation platform in human mitochondria (Reinhard et al, 2017). In this model, most, if not all, tRNAs are recognized and bound by the TRMT10C/SDR5C1 complex, presumably co-transcriptionally, and then act as a maturation platform to stabilize the partially degenerate human mitochondrial tRNA pool during the ensuing maturation (Reinhard et al, 2017).

After 5′- and 3′-processing, mitochondrial tRNA maturation continues with the addition of a CCA triplet by the CCA-adding enzyme at the 3′-end (Rossmanith et al, 1995), presumably while still bound to the TRMT10C/SDR5C1 tRNA maturation platform. Interestingly, the 3′-CCA is an RNase Z antideterminant, which is recognized and not removed by ELAC2 to avoid futile cycling of the tRNAs between the CCA-adding enzyme and RNase Z (Mohan et al, 1999; Nashimoto, 1997). The 3′-CCA antideterminant effect ensures an efficient tRNA maturation process – the lack of which leads to a spectrum of diseases similar to those observed when CCA addition is impaired (Wedatilake et al, 2016).

Despite the importance of ELAC2 functionality for human health, the molecular basis of tRNA recognition, catalysis, and CCA antidetermination remains unknown. Here, we determined the structures of several ternary complexes of human mitochondrial RNase Z. Our structures reveal the molecular details of ELAC2's interactions with its tRNA substrate and the TRMT10C/SDR5C1 tRNA maturation platform. Furthermore, we show how ELAC2 discriminates against tRNAs with an unprocessed 5′-end or those bearing a 3′-CCA tail and provide a molecular rationale for several clinically relevant mutations.

## Results

### Cryo-EM structure of the mitochondrial RNase Z product ternary complex

To understand the structural basis of mitochondrial tRNA 3′-processing, we reconstituted the human mitochondrial RNase Z complex using recombinantly expressed TRMT10C–SDR5C1 subcomplex, a tRNA precursor, and ELAC2 (Fig. 1A; Appendix Fig. S1A). As a precursor tRNA substrate, we used the tRNA cluster tRNA$^{His}$-tRNA$^{Ser(AGY)}$ (abbreviated as HS) located on the mtDNA heavy strand (Fig. 1B). The recombinant ELAC2 is catalytically active, and the TRMT10C–SDR5C1 subcomplex significantly accelerated the processing by ELAC2 (Fig. EV1), as previously shown (Reinhard et al, 2017). The reconstituted catalytically active RNase Z (Appendix Fig. S1B) was incubated to complete turnover and used to determine a cryo-EM structure of the ternary product complex of human RNase Z to an overall resolution of 2.9 Å (Fig. 1C,D; Appendix Figs. S2 and S3; Table 1). Within the complex, a tetramer of SDR5C1 forms a flat platform that can be bound from each flat side by TRMT10C and precursor mt-tRNA. Thus, the SDR5C1/TRMT10C subcomplex can support the simultaneous processing of two tRNA molecules. As the SDR5C1 platform has twofold symmetry within the plane, we observed TRMT10C/tRNA in either of the two equivalent positions rotated by 180° to each other (Fig. EV1B).

The SDR5C1/TRMT10C/tRNA part of the RNase Z product complex is similar to a previously published mt-RNase P structure, in which the SDR5C1/TRMT10C subcomplex interacts with PRORP to catalyze the 5′-cleavage of a precursor mt-tRNA (Bhatta et al, 2021). This highlights the notion that mt-tRNA processing by mt-RNase P and mt-RNase Z is tightly coupled (Reinhard et al, 2017) as it is only required to exchange the RNA processing catalytic subunit without major structural rearrangements in the SDR5C1/TRMT10C/tRNA platform. In our product complex, D314–P319 in TRMT10C, known as motif II in the TrmD-family of tRNA methyltransferases, have a different conformation compared to that observed in the apo-TRMT10C structure (Bhatta et al, 2021). Here, the motif II loop's conformation is similar to the SAM-bound TRMT10C methyltransferase (PDB: 5NFJ; Appendix Fig. S4).

Furthermore, the structure shows how TRMT10C compensates for the lack of strong D arm/T arm interactions in most bilaterian mitochondrial tRNAs by providing structural support through its extensive interactions with the mt-tRNA's D arm. In addition, TRMT10C's N-terminal domain stabilizes the T loop via electrostatic interactions with the RNA backbone (Fig. 2A–C). Collectively, these interactions underscore TRMT10C's importance in promoting bilaterian mt-tRNA folding, and these elements are recognized by ELAC2 when in the complex with the SDR5C1/TRMT10C tRNA processing platform.

The structure of human ELAC2 is similar to that of the yeast homolog Trz1. It consists of two β-lactamase domains (N-terminal domain (NTD) and C-terminal domain (CTD)) with insertions and extensions (Fig. 1A). Although the NTD and CTD have the same fold (RMSD of 2.1 Å for 212 Cα atoms), the NTD lacks the key Zn$^{2+}$-bound catalytic site necessary for hydrolytic activity. In addition, similar to Trz1, the human ELAC2 active site on the CTD is situated in a channel formed between the NTD and the CTD (Fig. 1C,D).

In the mitochondrial RNase Z complex, ELAC2 binds to the mt-tRNA T loop and the TRMT10C N-terminal domain through the exosite (Fig. 3A,B). The exosite establishes electrostatic interactions with the T loop backbone U55 via K279 and N253. Furthermore, the exosite interacts with and is stabilized in its position by the TRMT10C N-terminal domain. Here, ELAC2 residues V256, L257, and the alkyl chain of K260 form hydrophobic interactions with the tip of the TRMT10C N-terminal α-helix, including L103 and L104. The hydrophobic character of these ELAC2 and TRMT10C positions is conserved in mammals and, therefore, their capacity to form a hydrophobic zipper that links ELAC2 to the SDR5C1/TRMT10C tRNA processing platform (Appendix Fig. S5A,B) to form a clamp over the tRNA T loop (Fig. 3B). To test the importance of this ELAC2-TRMT10C hydrophobic interaction, we assayed the activity of a V256E/L257E mutant. We found that the activity of the ELAC2 V256E/L257E mutant in the presence of TRMT10C/SDR5C1 is comparable to that of wild-type ELAC2 in the absence of TRMT10C/SDR5C1 (Fig. EV2A). Notably, the TRMT10C N-terminal α-helix is also strongly contributing to the formation of the mitochondrial RNase P complex through clamping the tRNA T loop and PRORP (Fig. EV2B).

The ELAC2 C-terminal α-helix (α25), which is only present in long-form homologs (Appendix Fig. S5C), contributes further to mitochondrial RNase Z complex formation (Fig. 3C). The C-terminal helix contains R788, which interacts with the backbone of U69, located close to the tip of the acceptor stem. Furthermore, R788, along with R791, binds Q343 in TRMT10C (Fig. 3C). ELAC2 R788 is conserved in metazoans while R791 is conserved in

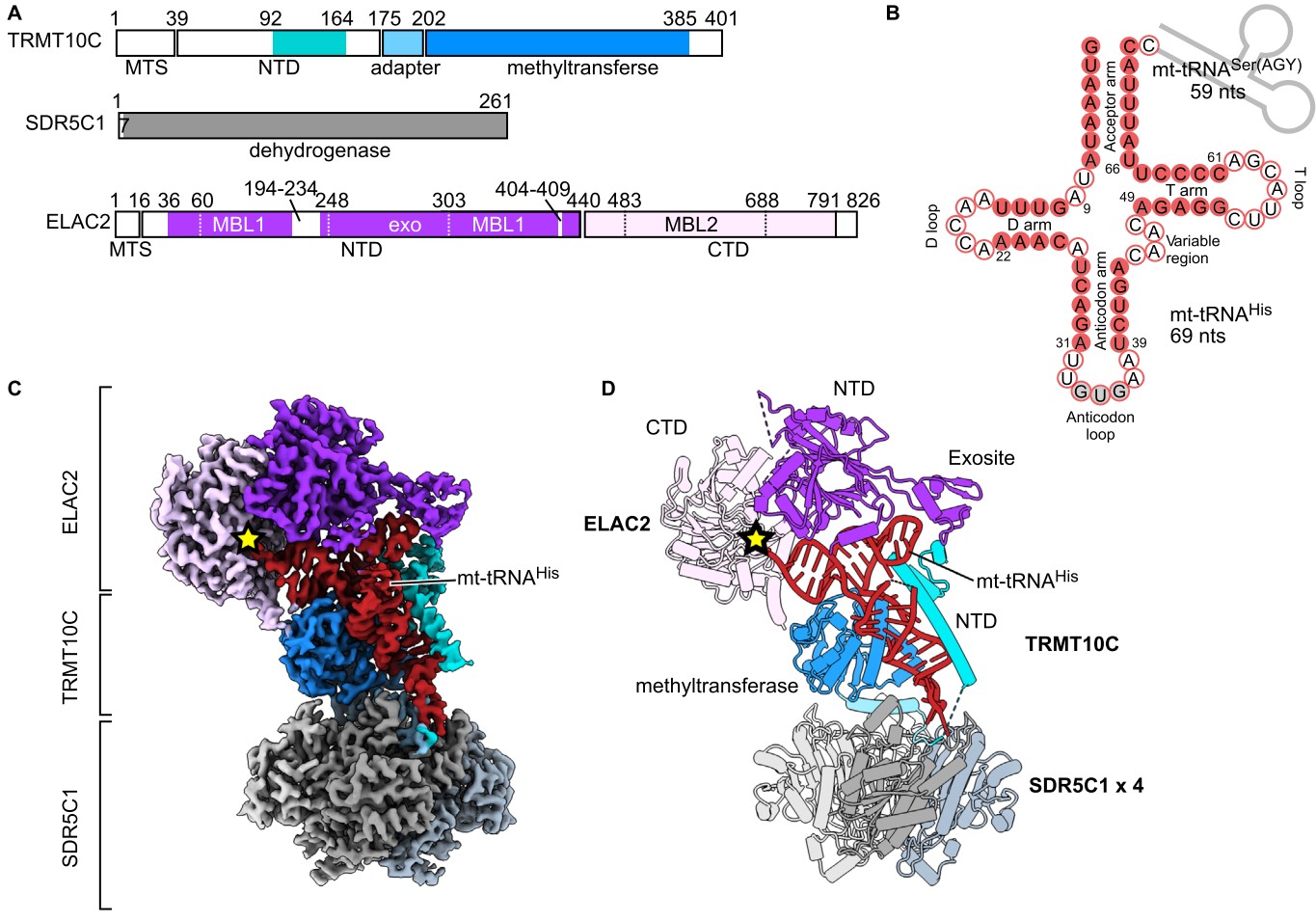

**Figure 1. Structure of human mitochondrial RNase Z.**

(A) Domain architecture of the human mitochondrial RNase Z components. The N-terminal (NTD) and C-terminal (CTD) domains of ELAC2 are shown in dark and light purple, respectively. MBL1 and MBL2, which are the metallo-β-lactamase folds, and the exosite insertion are indicated with dotted lines. TRMT10C is shown in blue, with the NTD highlighted in cyan. MTS, mitochondrial targeting sequence. Unmodelled regions are in white. (B) The mt-tRNA precursor consists of tRNA$^{His}$ (red) and tRNA$^{Ser(AGY)}$ (gray line), of which the latter has been cleaved off in the ternary product structure. Base-paired nucleotides are in red circles, and non-base-paired ones are in white circles. The anticodon triplet is in gray circles. Canonical tRNA numbering is indicated (Giegé et al, 2012) and is used throughout the text (the correspondence with nucleotide numbers in mitochondrial tRNA$^{His}$, which was used in the PDB deposited structures, is shown in Appendix Table S1). (C, D) Composite cryo-EM density (C) and cartoon representation (D) of the RNase Z complex. Colored as in A. The four SDR5C1 subunits are shown in different shades of gray. The ELAC2 active site is marked with a yellow star.

mammals (Appendix Fig. S5C). TRMT10C Q343 is partially conserved in metazoans (Appendix Fig. S5D). The C-terminal α helix sterically hinders a 5′-extension of the mt-tRNA (Fig. EV2C) and, therefore, provides the structural basis for the 5′-to-3′ processing order observed in vivo and in vitro (Rackham et al, 2016; Reinhard et al, 2017; Rossmanith et al, 1995). In addition, ELAC2 residues R381, K495, and K731 interact with the 5′-α-phosphate of the mt-tRNA$^{His}$ (Fig. 3D). Thus, these three residues may probe if correct 5′-processing has been performed previously by RNase P through electrostatic interactions with the 5′-monophosphate group.

ELAC2 establishes sequence-independent electrostatic interactions with the major groove between the acceptor stem and the T loop (Fig. 3E). The conserved lysines and arginine in the 99-KLKVAR-104 motif (Appendix Fig. S5E) align well to interact with three consecutive backbone phosphates. Furthermore, two conserved residues,

R38 and R41 (Appendix Fig. S5F), in the ELAC2 N-terminal α helix (α1) line up with and assist the 99-KLKVAR-104 motif in acceptor-arm recognition (Fig. 3E). In total, ELAC2 possesses five positively charged side chains that behave like a zipper to anchor the enzyme on the bound mt-tRNA.

Taken together, these results show that ELAC2 recognizes the basic structural features of mt-tRNAs. In addition, ELAC2 is held in place and straddled over the tRNA 3′-end by its N- and C-terminal interactions with TRMT10C on the TRMT10C/SDR5C1 tRNA maturation platform (Fig. 3).

## Structural basis of human mitochondrial RNase Z catalysis

In our product complex RNase Z-HS, the discriminator base C73 is positioned in the active site, with its ribose-O3 in the expected position

**Table 1. Cryo-EM data collection, refinement, and validation statistics.**

| | RNase Z-HS (EMDB-50050) (PDB 9EY0) | RNase Z$^{H548A}$-HS (EMDB-50051) (PDB 9EY1) | RNase Z-HCCA (EMDB-50052) (PDB 9EY2) | RNase Z-HCCA TRMT10C/SDR5C1 focus (EMDB-51230) (PDB 9GCH) |
|---|---|---|---|---|
| **Data collection and processing** | | | | |
| Magnification | 165,000 | 165,000 | 165,000 | 165,000 |
| Voltage (kV), source | 300, XFEG | 300, XFEG | 300, E-CFEG | 300, E-CFEG |
| Electron exposure (e$^-$/Å$^2$) | 59 | 59 | 59 | 59 |
| Underfocus range (μm) | 0.2–2 | 0.2–2 | 0.2–2 | 0.2–2 |
| Pixel size (Å) | 0.505 | 0.505 | 0.505 | 0.505 |
| Symmetry imposed | P1 | P1 | P1 | P1 |
| Initial particle images (no.) | 293,145 | 643,629 | 4,773,315 | 4,773,315 |
| Final particle images (no.) | 28,771 | 15,961 | 4346 | 293,773 |
| Map resolution (Å)[a] | 2.94 | 2.92 | 2.96 | 1.90 |
| Global refinement | 2.78[b] | 2.69[b] | 3.1[c] | |
| ELAC2 focus | 3.07[b] | 3.34[b] | 3.2[c] | |
| FSC threshold | 0.143 | 0.143 | 0.143 | 0.143 |
| **Map resolution range (Å)[d]** | | | | |
| Min, 25th percentile | 2.204, 3.110, 3.872 | 2.204, 3.153 | 1.486, 3.372 | 1.545, 2.007 |
| Median, 75th percentile | 6.156, 44.472 | 3.845, 5.530, 46.799 | 4.560, 7.171, | 2.392, 3.750, 26.869 |
| Max | | | 48.229 | |
| **Refinement** | | | | |
| Initial model used (PDB code) | 7ONU | 7ONU | 7ONU | 7ONU |
| Model resolution (Å) | 3.0 | 2.8 | 3.1 | 1.9 |
| FSC threshold | 0.5 | 0.5 | 0.5 | 0.5 |
| **Model composition** | | | | |
| Non-hydrogen atoms | 16,691 | 16,783 | 16,556 | 11,235 |
| Protein/RNA residues | 2008/65 | 2009/69 | 1982/68 | 1304/68 |
| Ligands | GTP:1 Zn:2 SAM:1 | GTP:1 Zn:1 SAM:1 | GTP:1 Zn:2 SAM:1 | GTP:1 Mg:1 SAM:1 |
| **$B$ factors (Å$^2$)** | | | | |
| Protein | 22.1/186.4/74.99 | 0/106.2/35.1 | 24.69/419.90/52.20 | 2.08/144.89/25.72 |
| RNA | 40.5/147.0/90.5 | 9.8/167.9/70.0 | 28.66/141.64/58.40 | 5.91/158.23/44.49 |
| Ligand | 42.8/189.2/85.3 | 78.55/171.39/91.80 | 35.69/164.93/76.06 | 13.90/145.87/70.29 |
| Water | | | | 18.21/34.63/26.14 |
| **r.m.s. deviations** | | | | |
| Bond lengths (Å) | 0.004 | 0.005 | 0.004 | 0.003 |
| Bond angles (°) | 0.540 | 0.634 | 0.504 | 0.489 |
| **Validation** | | | | |
| MolProbity score | 1.36 | 1.38 | 1.38 | 1.19 |
| Clashscore | 4.49 | 4.90 | 6.16 | 4.05 |
| Poor rotamers (%) | 0.18 | 0.43 | 0.00 | 0.00 |
| **Ramachandran plot** | | | | |
| Favored (%) | 97.29 | 97.34 | 97.81 | 98.22 |
| Allowed (%) | 2.71 | 2.66 | 2.09 | 1.78 |
| Disallowed (%) | 0.00 | 0.05 | 0.10 | 0.00 |

[a]Gold-standard 0.143 FSC resolution calculated with 3DFSC (Tan et al, 2017). The masks for FSC calculation were generated in CryoSPARC using a relative threshold of 0.5.
[b]Gold-standard 0.143 FSC resolution calculated by CryoSPARC with the auto-tightened refinement mask.
[c]Gold-standard 0.143 FSC resolution calculated by RELION postprocess using the refinement mask.
[d]Map resolution range calculated with CryoSPARC v4.3.1 BlocRes and FSC threshold of 0.5.

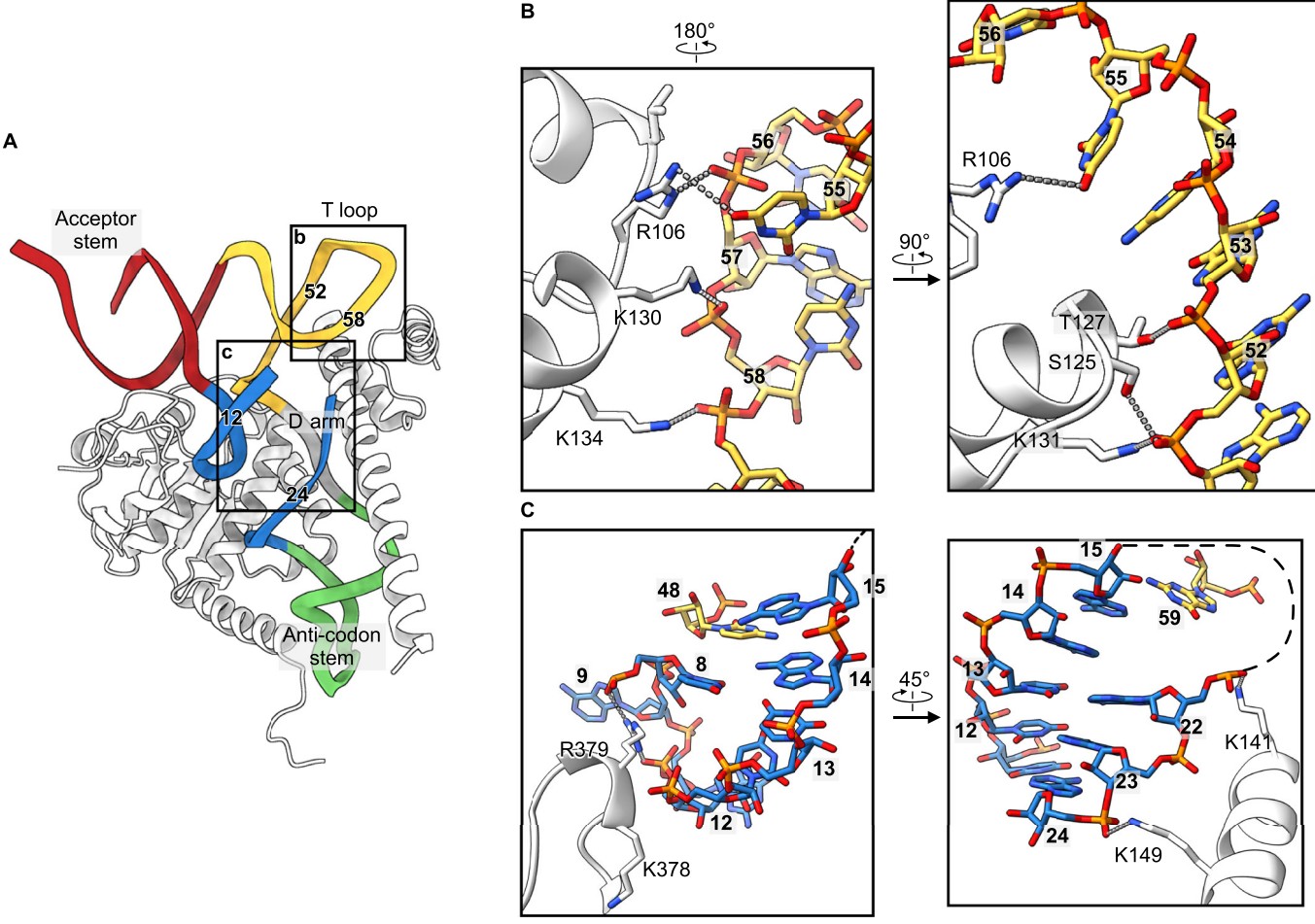

**Figure 2. Stabilization of the tRNA fold by TRMT10C.**

(A) TRMT10C (white) wraps around the mt-tRNA^His anticodon stem and D arm to stabilize the cloverleaf fold. The nucleotide numbers forming the T loop are indicated for comparison with those in (B, C). (B, C) Molecular interactions of TRMT10C with the T loop (B) and the D arm (C). The views have been rotated for easier visualization.

directly after catalysis (Fig. 4A). Furthermore, the C73 backbone phosphate is wedged between S490, S768, and K700, whereas R728 in the conserved 724-HFSQRY-729 motif (Appendix Fig. S5G) clamps the C72 and C73 backbone phosphates (Fig. 4A). Several clinically relevant mutations associated with hypertrophic cardiomyopathy have been identified in human patients. In particular, three of them affect the substrate-binding residues in and surrounding the HFSQRY motif and reduce the $k_{cat}/K_M$ of ELAC2 (Saoura et al, 2019). A clinically relevant mutation in the HFSQRY motif is Y729C. Here, Y729 stacks on R728 and steers R728 towards the precursor 3′-end (Fig. 4A). Another clinically relevant mutation in relation to the HFSQRY motif is P493L. Here, Q727 in the HFSQRY motif is strongly held (2.2 Å H-bond) in place by the backbone carbonyl of I492. A P493L mutation will significantly affect this interaction, thus preventing Q727 from remaining anchored as needed to ensure that R728 is properly aligned (Fig. 4A). The third clinically relevant mutation related to the HFSQRY motif is R781H. Here, R781 binds and fixates the R728 backbone carbonyl, which is crucial for R728 to be properly oriented (Fig. 4A).

To trap an mt-RNase Z substrate-relevant complex, we mutated H548, which coordinates one of the two active site $Zn^{2+}$ ions and determined a 2.9 Å resolution cryo-EM structure from a

mitochondrial RNase Z (H548A) complex preparation in which the mt-tRNA precursor was predominantly uncleaved (RNase Z^H548A-HS; Fig. 4B,C; Appendix Figs. S1B, S6, S7; Table 1). Overall, the structure of the TRMT10C/SDR5C1/mt-tRNA platform did not change with the introduction of the H548A mutation. However, ELAC2 pivots on the mt-tRNA by approximately 3.5° (Fig. EV3A) compared with its position in the product complex. Due to this pivoting, the RNase Z^H548A-HS structure shows a difference in the interactions around the discriminator nucleotide and the scissile bond (Fig. 4C) compared with the product complex. In the RNase Z^H548A-HS structure, the 3′-oxygen of the scissile bond is properly oriented for hydrolysis, but it is displaced by ~3 Å, and the discriminator nucleotide C73 moves closer to E130. This may be attributed to the absence of the first $Zn^{2+}$ ion due to the H548A mutation. Notably, K700 switches from interacting with C72 in the product complex RNase Z-HS to interacting with the scissile bond, i.e., the C73-G1 phosphodiester bond in the RNase Z^H548A-HS structure (Fig. 4C).

The scissile bond is followed by the mt-tRNA^Ser(AGY) acceptor stem. Only the first nucleotide (G1) was modeled; nevertheless, the cryo-EM map contains additional density for the mt-tRNA^Ser(AGY)

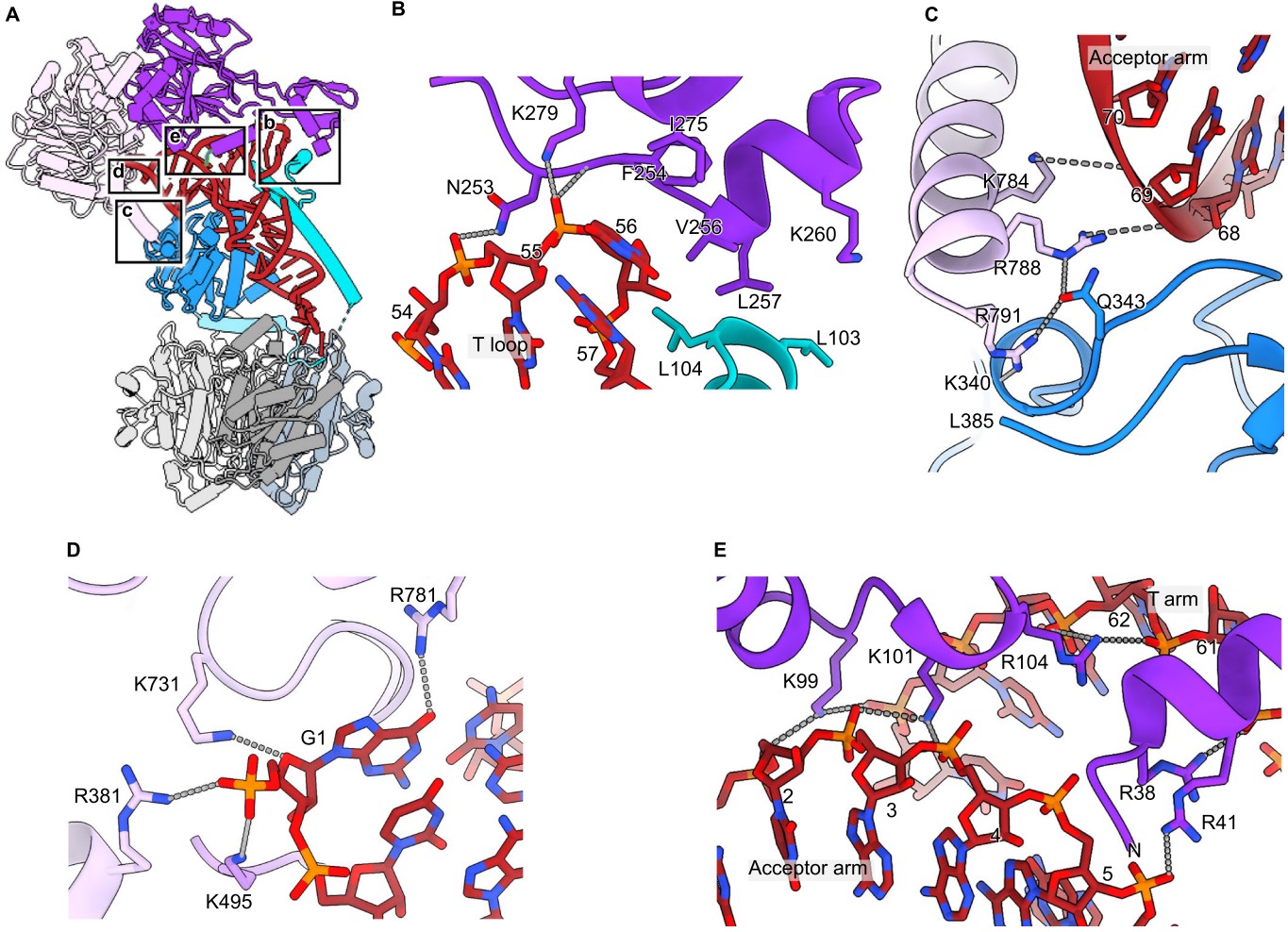

**Figure 3. Interaction of ELAC2 with the precursor mt-tRNA^His.**

(A) Overview of the human mitochondrial RNase Z complex. The boxes represent the insets for (B–E). (B) Contacts between the ELAC2 exosite, the T loop, and the TRMT10C N-terminal domain. (C) Contacts between the ELAC2 C-terminal helix, the TRMT10C, and the acceptor stem. (D) Recognition of the 5′-end of the mt-tRNA. Only the α-phosphate of G1 is shown since natural ELAC2 substrates carry a 5′-monophosphate. (E) Interaction of ELAC2 with the acceptor stem/T arm groove.

(Figs. 4B and EV3B) and reveals that the acceptor stem contacts ELAC2 residues G156-P157 (Fig. EV3C), which are situated on a ridge in the 3′-direction of the substrate RNA. The peptide backbone undergoes a conformational change, which might be caused by the RNA contact. This suggests that G156 and P157 are involved in substrate binding, and this is further supported by the clinically relevant F154L mutation since F154 plays a structural role in the stabilization of this ridge (Saoura et al, 2019).

Our RNase Z^H548A-HS structure is comparable to the X-ray structure of *B. subtilis* RNase Z obtained using a non-cleavable substrate analog (Pellegrini et al, 2012). The human ELAC2 G1 corresponds to U1 in *B. subtilis*. The U1 inserts into a pocket between the two β-lactamase domains, and this interaction is necessary for hydrolysis. In human ELAC2, G1 is sterically occluded from the pocket by L126. This indicates that, unlike in *B. subtilis*, the pocket is not used to process the mt-tRNA^His-Ser(AGY) precursor (Fig. EV3D).

Taken together, in comparison with the RNase Z-HS product structure, the RNase Z^H548A-HS structure reveals additional ELAC2

residues involved in binding to the mt-tRNA precursor and in orienting the scissile bond, namely E130, K700, and G156-P157.

## Structural basis for the 3′-CCA tail as an antideterminant for ELAC2 processing

After maturation of the mt-tRNA ends by RNase P and RNase Z, the mitochondrial CCA-adding enzyme adds the 3′-terminal cytosine-cytosine-adenine (3′-CCA) triplet. The 3′-CCA is necessary for the aminoacylation of tRNAs, and thus, it should be protected from ELAC2 activity to prevent futile cycles of addition and removal. Indeed, ELAC2 activity on cytosolic tRNAs bearing the 3′-CCA tail is very inefficient (Mohan et al, 1999; Nashimoto, 1997).

To evaluate the antideterminant effect of the 3′-CCA tail in the mitochondria, mt-tRNA^His carrying a 3′-CCA tail was incubated with TRMT10C/SDR5C1 and increasing concentrations of ELAC2. The removal of the 3′-CCA tail by ELAC2 was very inefficient compared to the precursor trailer GAG of mt-tRNA^His (Fig. 5A).

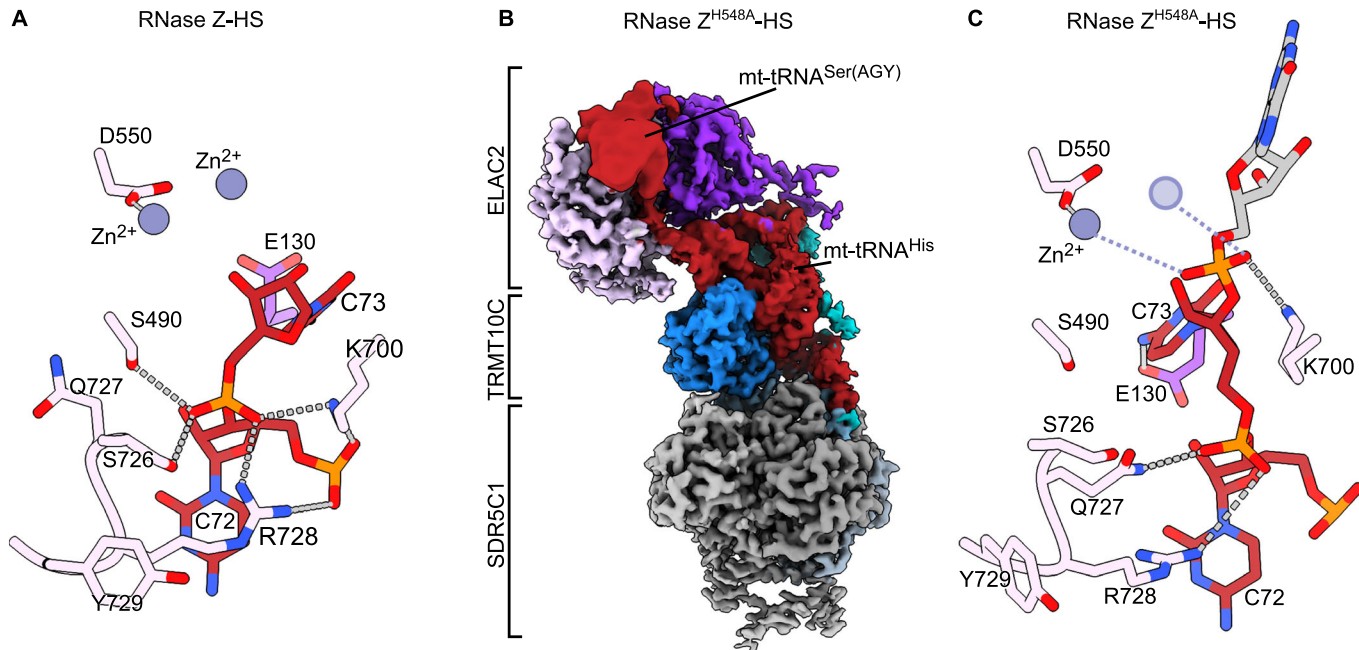

**Figure 4. Interactions around the tRNA scissile bond and the ELAC2 HFSQRY motif.**

(A) In the RNase Z-HS structure, the 3'-oxygen interacts with a $Zn^{2+}$ after the 3'-trailer is removed (B) Composite cryo-EM density of the human mitochondrial RNase Z bound to the mt-tRNA$^{His-Ser(AGY)}$ precursor, carrying the ELAC2 mutation H548A. The part of the map corresponding to mt-tRNA$^{Ser(AGY)}$ moiety was gaussian-filtered with a B-factor of 600 in UCSF ChimeraX. Colored as in Fig. 1. (C) The RNase Z$^{H548A}$-HS structure shows how the scissile bond is well-oriented for catalysis assisted by K700. The missing $Zn^{2+}$ due to the H548A mutation is shown by a transparent circle based on the RNase Z-HS structure. In the RNase Z-HS structure, the 3'-oxygen interacts with a $Zn^{2+}$ after the 3'-trailer is removed.

To understand the molecular details of 3'-CCA discrimination, we determined the cryo-EM structure to 3.0 Å resolution of RNase Z with mt-tRNA$^{His}$ carrying the 3'-CCA tail (RNase Z-HCCA, Fig. 5B, Appendix Figs. S8–S10; Table 1).

The TRMT10C/SDR5C1 platform in the RNase Z-HCCA complex is similar to the RNase Z-HS and RNase Z$^{H548A}$-HS complexes, but here, ELAC2 pivots by ~6 degrees around the tRNA (Fig. EV3A). Consequently, the active site is now ~13 Å away from the mt-tRNA$^{His}$ 3'-end, and the active site channel is occupied by the 3'-CCA tail (Fig. 5C).

The first C (C1$^{CCA}$) of the 3'-CCA tail is recognized by H-bonding to S726 and S490 and to the backbone carbonyls 490 and 492 (Fig. 5D), which lie at the entrance of the active site channel. The second C (C2$^{CCA}$) inserts into a pocket in the NTD situated on the opposite side of the active site channel. C2$^{CCA}$ establishes H-bonds with Q92 and T127, which pulls C2$^{CCA}$ toward the bottom of the pocket (Fig. 5E). Furthermore, the phosphodiester bond between C1$^{CCA}$ and C2$^{CCA}$ is stabilized by K700 (Fig. 5C). The terminal A (A3$^{CCA}$) base in the CCA does not establish H-bonds. It, therefore, does not seem to participate in the antideterminant effect in accordance with previous in vitro work (Nashimoto, 1997).

The C1$^{CCA}$–C2$^{CCA}$ bond is the closest to the active site. Nevertheless, it is not well positioned for hydrolysis since it is 1.9 Å further away from the active site than the scissile bond in the mt-tRNA$^{His-Ser(AGY)}$ complex. Furthermore, the phosphate oxygens point away from the active site $Zn^{2+}$, so they cannot participate in the hydrolysis reaction catalyzed by ELAC2 (Fig. 5F). This

conformation of the RNA backbone is enforced by the interactions established by both C1 and C2.

Most of the residues that interact with the 3'-CCA tail are also relevant for catalysis (Fig. EV4A). To identify residues solely important for CCA tail recognition, we selected three clinically relevant mutations that are structurally close to the HFSQRY motif but do not significantly alter the catalytic properties of ELAC2 (Haack et al, 2013; Paucar et al, 2018; Saoura et al, 2019). These are G132R, F154L and T520I (Fig. EV4B). None of these mutations interferes significantly with the binding of the 3'-CCA tail since the 3'-CCA antidetermination is not uncoupled to the effect on catalysis (Fig. EV4C). This strengthens the notion that, in ELAC2, 3'-trailer cleavage and 3'-CCA antidetermination depend on highly interdependent mechanisms.

## Discussion

The structural analysis presented here reveals how ELAC2 interacts with the tRNA processing platform formed by TRMT10C/SDR5C1 to form human mitochondrial RNase Z.

The binding of ELAC2 through both its N-terminal and C-terminal domains to TRMT10C underscores a complex and finely tuned interaction network essential for the precise positioning and enzymatic activity of ELAC2 on its tRNA substrate. ELAC2 straddling the tRNA molecule, facilitated by its interactions at both terminal domains, exemplifies an indirect readout mechanism in which the enzyme recognizes and interacts with the structural features of tRNA

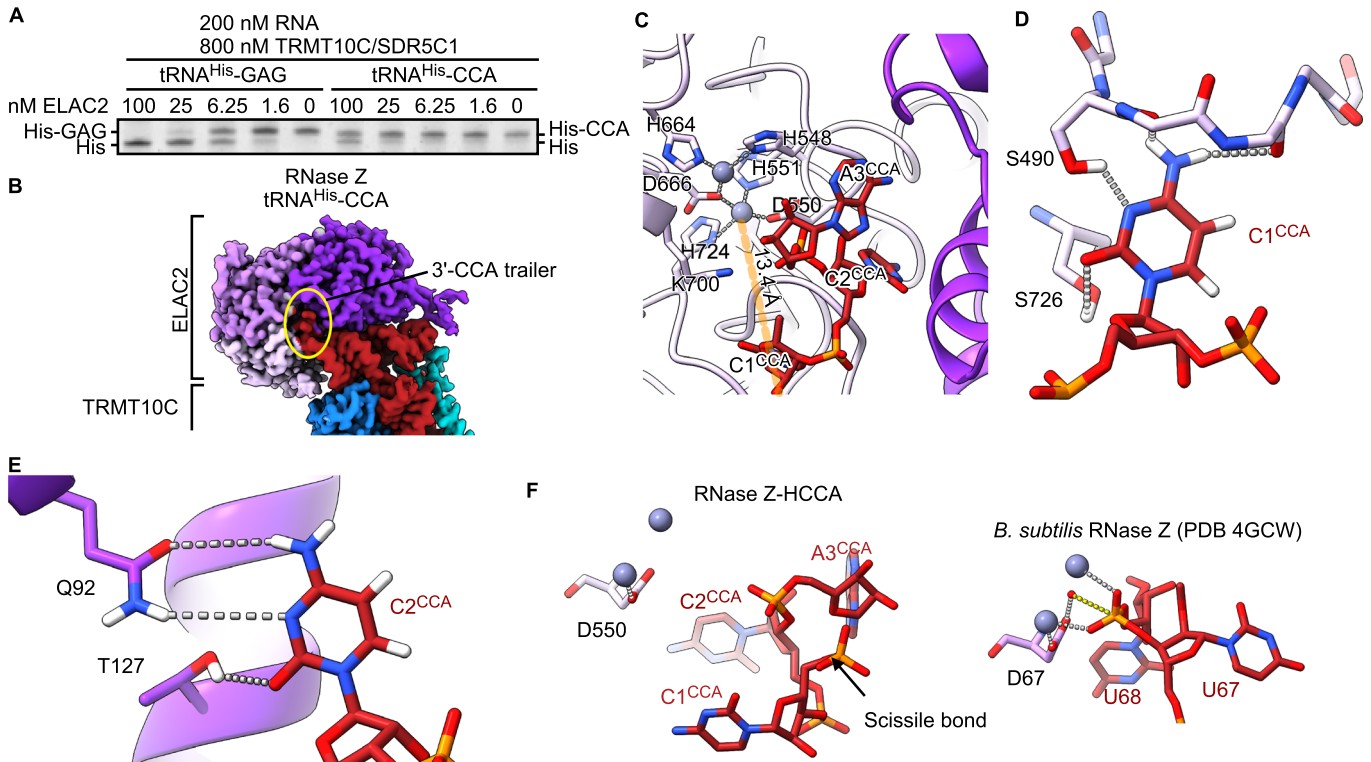

**Figure 5. Structural basis for 3′-CCA discrimination.**

(A) ELAC2 dilution series on 200 nM mt-tRNA$^{His}$-GAG or mt-tRNA$^{His}$-CCA and 800 nM TRMT10C/SDR5C1 was analyzed using a TBE-UREA PAGE. Representative gel image of technical triplicates and biological duplicates for WT ELAC2. (B) Composite cryo-EM density of human mitochondrial RNase Z bound to the mt-tRNA$^{His}$-CCA. Colored as in Fig. 1. The density for the 3′-CCA trailer is indicated with a yellow circle. (C) The 3′-CCA occupies the active site channel of ELAC2. The mt-tRNA$^{His}$ 3′-end is 13.4 Å away from the catalytic Zn$^{2+}$ (gray spheres). (D) Interactions of C1$^{CCA}$ with ELAC2 S490 and S726. (E) Interactions of C2$^{CCA}$ with Q92 and T127. The hydrogen atoms were added in ChimeraX to visualize the H-bonding interactions. (F) Comparison with the tRNA$^{Thr}$ precursor bound to the B. subtilis RNase Z (Pellegrini et al, 2012), which shows the interactions required for catalysis. The phosphate oxygens coordinate one Zn$^{2+}$ ion each. The hydrolyzing water molecule (red sphere) is positioned by D550. In RNase Z-HCCA, the backbone adopts a different conformation, which is unsuitable for catalysis. Source data are available online for this figure.

rather than directly binding to specific nucleotide sequences. This mode of recognition allows ELAC2 to effectively process a variety of tRNAs by accommodating variations within the tRNA structure, thus maintaining a high degree of processing fidelity across different tRNA species. The dual interaction sites on ELAC2 enhance its stability and positioning on the tRNA-bound TRMT10C/SDR5C1 platform, aligning ELAC2 optimally for catalysis. This structural arrangement is crucial for the enzymatic function of ELAC2, as it ensures that the enzyme is correctly oriented relative to the tRNA substrate. The N-terminal interaction likely contributes to the initial recognition and binding of the tRNA, positioning ELAC2 to engage with the substrate. Subsequently, the C-terminal interaction stabilizes the complex during the catalytic process, ensuring efficient and precise cleavage at the 3′-end of the tRNA. Moreover, the strategic positioning of ELAC2 over the tRNA, secured by interactions at both terminals, indicates a robust regulatory mechanism. This setup facilitates the catalytic action required for tRNA maturation and prevents aberrant processing that could lead to dysfunctional tRNA molecules. Thus, the interaction between ELAC2 and the tRNA maturation platform reflects a highly evolved and specialized system designed to meet the unique demands of mitochondrial biology, where precision in tRNA processing is directly linked to the organelle's efficiency in protein synthesis.

We have previously shown that the TRMT10C/SDR5C1 complex enhances ELAC2 activity on most mt-tRNA precursors, with the exceptions of mt-tRNA$^{Gln}$, mt-tRNA$^{Asn}$, mt-tRNA$^{Leu(UUR)}$, which belong to Type 0 (canonical) tRNAs, and mt-tRNA$^{Ser(UCN)}$, which belongs to Type I (Helm et al, 2000; Reinhard et al, 2017). They all contain the UU(C/U) sequence in the T loop and the GG sequence in the D loop. Also belonging to Type 0 are nuclear-encoded tRNAs, which ELAC2 processes in the nucleus (Siira et al, 2018) without TRMT10C/SDR5C1. These features of Types 0 and I enable the base pairing of the D and T loops and are, therefore, important for the stability of the cloverleaf fold.

In contrast, the other mt-tRNAs, which are processed more efficiently in the presence of TRMT10C/SDR5C1, belong to type II tRNAs (Watanabe, 2010), which lack the aforementioned D loop-T loop features (Helm et al, 2000; Wakita et al, 1994). Consequently, the folding of the T loop is less stable in type II tRNAs. Here, the structures show that the NTD of TRMT10C binds to the tRNA T loop to stabilize its fold and facilitate the tRNA interaction with the ELAC2 exosite. Furthermore, the TRMT10C/SDR5C1 tRNA maturation platform's role extends beyond mere structural support; it also likely impacts the kinetic properties of the processing enzymes. By pre-organizing the tRNA substrates, the platform may reduce the entropy cost during the transition state of the enzymatic

reaction, thus enhancing the catalytic efficiency. This pre-organization is crucial in mitochondria, where the compact genome and the need for rapid and accurate protein synthesis necessitate highly efficient processing mechanisms.

The observations above underscore the role of TRMT10C/SDR5C1 as a platform for mt-tRNA processing, and this is further supported by other research made available during the preparation of this manuscript (preprint: Bhatta et al, 2024; Meynier et al, 2024). As such, it supports the multistep processing of mt-tRNA precursors from 5′-endonucleolytic cleavage by PRORP, 3′-endonucleolytic cleavage by ELAC2, and 3′-CCA addition. These steps occur in strict sequential order (Lopez Sanchez et al, 2011; Rackham et al, 2016; Reinhard et al, 2017; Rossmanith et al, 1995). Our structure of RNase Z shows that the ELAC2 C-terminal helix (α24) would sterically clash with 5′-unprocessed pre-tRNAs (Fig. EV2C) and, therefore, force ELAC2 to act after RNase P. In addition, ELAC2 interactions with the 5′-phosphate via R381, K495, and K731 are important for positioning ELAC2 for catalysis.

After 3′-cleavage by ELAC2, the 3′-CCA tail is added to mt-tRNA precursors, which is required for aminoacylation. To prevent futile cycles of 3′-CCA tail addition and removal, ELAC2 recognizes and ignores precursor tRNAs that carry the 3′-CCA tail (Mohan et al, 1999; Nashimoto, 1997). In particular, (i) the first nucleotide must be a C for the strongest antideterminant effect; (ii) a C or U in the second position enhances the antideterminant effect; (iii) the third position does not contribute to the antideterminant effect and can be any nucleotide. Our RNase Z-HCCA structure shows that ELAC2 recognizes the first two "C" nucleotides in the 3′-CCA tail: $C1^{CCA}$ and $C2^{CCA}$. Based on our structure, purines are likely sterically occluded from the $C1^{CCA}$-binding site, whereas uracil, on the other hand, is disfavored due to a mismatching H-bond pattern (Fig. 5D). Indeed, the mutation of 3′-UCU to 3′-CCU in mt-tRNA$^{Ser(UCN)}$ generates a 3′-trailer that cannot be processed by mitochondrial RNase Z and is a cause of non-syndromic deafness (Levinger et al, 2001). Furthermore, $C2^{CCA}$ inserts into a pocket in the NTD of ELAC2, where it interacts with Q92. This interaction pulls the C1–C2 phosphodiester bond away from the active site, thereby preventing hydrolysis.

The role of the HFSQRY motif in ELAC2 presents a fascinating example of multifunctionality within mitochondrial tRNA processing enzymes. This motif is pivotal not only for the catalytic activity of ELAC2 but also plays a crucial role in recognizing the CCA end of tRNAs, which act as an antideterminant for ELAC2 catalysis. Therefore, mutations in the HFSQRY motif can impact both the catalytic activity of ELAC2 and its ability to recognize the CCA sequence correctly; hence, it is thus difficult to separate these two effects (Fig. EV4B,C).

In summary, our study is the first to elucidate the complete structural framework of the human mitochondrial RNase Z complex in various maturation states of mitochondrial tRNA. By resolving the cryo-EM structures at molecular resolution, we have uncovered the molecular basis for the 5′-to-3′ tRNA processing order in mitochondria, the detailed mechanisms behind the 3′-CCA antideterminant effect, and the sequence-independent recognition of mitochondrial tRNA substrates. These findings significantly enhance our understanding of the mitochondrial tRNA maturation process, revealing the intricate interplay between different components of the RNase Z complex and their substrates. Furthermore, the study bridges a critical gap in linking specific mutations in

ELAC2 to mitochondrial pathologies, providing new insights into the molecular etiology of these diseases. The structural and mechanistic insights gained from our work offer a valuable framework for interpreting the pathogenicity of newly identified clinical mutations in the RNase Z complex and other associated factors. This improved understanding could help to more effectively diagnose, characterize, and potentially manage mitochondrial diseases associated with defects in tRNA processing.

# Methods

### Reagents and tools table

| Reagent/resource | Reference or source | Identifier or catalog number |
|---|---|---|
| **Experimental models** | | |
| *E. coli* KRX | Promega | L3002 |
| **Recombinant DNA** | | |
| pNIC-CTHF-M21 | Reinhard et al, 2017 | N/A |
| pJ411 | DNA TwoPointO | N/A |
| pJ411:His-TEV-HsELAC2 | DNA TwoPointO | N/A |
| pJ411::His-TEV-HsELAC2_H548A | This study | N/A |
| pJ411::His-TEV-HsELAC2_T520I | This study | N/A |
| pJ411::His-TEV-HsELAC2_F154A | This study | N/A |
| pJ411::His-TEV-HsELAC2_F154L | This study | N/A |
| pJ411::His-TEV-HsELAC2_G132R | This study | N/A |
| pJ411::His-TEV-HsELAC2_V256E_L257E | This study | N/A |
| Mitochondrial tRNAHis-Ser with 5′-T7 promoter (synthetic DNA; GBlock) | Integrated DNA Technologies | NCBI NC012920.1 nucleotides 12138-12265; Assembly GRCh38.p14) |
| **Oligonucleotides and other sequence-based reagents** | | |
| tRNAHis-GAG-rev | Integrated DNA Technologies | CTCGGTAAATAAGGGGTCGTAAGC |
| tRNAHis-CCA-rev | Integrated DNA Technologies | TGGGGTAAATAAGGGGTCGTAAGC |
| tRNAHis-for | Integrated DNA Technologies | CCGCGAATTAATACGACTCACTATAG |
| tRNA-Ser(AGY)-rev | Integrated DNA Technologies | TGAGAAAGCCATGTTGTTAGACATG |
| ELAC2_T520I_F | Integrated DNA Technologies | GCTGCTGGATTGTGGTGAGGGTATCTTCGGTCAACTGTGCCGCC |
| ELAC2_T520I_R | Integrated DNA Technologies | GGCGGCACAGTTGACCGAAGATACCCTCACCACAATCCAGCAGC |
| ELAC2_F154L_F | Integrated DNA Technologies | GAAATACTTGGAAGCCATTAAGATCCTGTCCGGTCCGCTGAAAGGCATC |
| ELAC2_F154L_R | Integrated DNA Technologies | GATGCCTTTCAGCGGACCGGACAGGATCTTAATGGCTTCCAAGTATTTC |
| ELAC2_F154A_F | Integrated DNA Technologies | GAAATACTTGGAAGCCATTAAGATCGCGTCCGGTCCGCTGAAAGGCATC |
| ELAC2_F154A_R | Integrated DNA Technologies | GATGCCTTTCAGCGGACCGGACGCGATCTTAATGGCTTCCAAGTATTTC |
| ELAC2_G132R_F | Integrated DNA Technologies | GATTCTGACCCTGAAAGAAACCCGTCTGCCGAAGTGCGTGCTGAGC |
| ELAC2_G132R_R | Integrated DNA Technologies | GCTCAGCACGCACTTCGGCAGACGGGTTTCTTTCAGGGTCAGAATC |

| Reagent/ resource | Reference or source | Identifier or catalog number |
| --- | --- | --- |
| | ELAC2_VL256EE_F | Integrated DNA Technologies |
| | | GCACCTGAAACGCGGCAATTTTCTGGAA GAAAAAGCGAAAGAGATGGGTCTGCCGG |
| | ELAC2_VL256EE_R | Integrated DNA Technologies |
| | | CCGGCAGACCCATCTCTTTCGCTTTTTCT TCCAGAAAATTGCCGCGTTTCAGGTGC |
| **Chemicals, enzymes, and other reagents** | | |
| DNase I | Roche | 4536282001 |
| HiPrep™ Sephacryl™ S-300 HR | Cytiva | Cytiva 17-1167-01 |
| HiTrap® Capto™ S | Cytiva | Cytiva 29-4004-58 |
| HiTrap® Heparin High Performance | Cytiva | Cytiva 17-0406-01 |
| HiTrap® Q High Performance | Cytiva | Cytiva 17-1153-01 |
| HiTrap™ IMAC FF | Cytiva | Cytiva 17-0921-02 |
| Mono Q® 5/50 GL | Cytiva | Cytiva 17-5166-01 |
| Novex™ TBE-Urea Gels, 15% | Thermo Fisher Scientific | EC6885BOX |
| Phusion™ High-Fidelity DNA Polymerase | Thermo Fisher Scientific | F530S |
| Proteinase K Solution (20 mg/mL), RNA grade | Thermo Fisher Scientific | 25530049 |
| Superdex® 200 Increase 10/300 GL | Cytiva | Cytiva 28-9909-44, |
| SYBR™ Green II RNA Gel Stain | Thermo Fisher Scientific | S7568 |
| T7 RNA polymerase | In-house production | N/A |
| TEV protease | In-house production | N/A |
| UltrAuFoil 300 mesh R 1.2/1.3 geometry | Quantifoil Micro Tools | Q350AR13A |
| **Software** | | |
| AlphaFold | Jumper et al, 2021 | N/A |
| Coot | Emsley et al, 2010 | N/A |
| CryoSPARC | Punjani et al, 2017 | N/A |
| EMReady | He et al, 2023 | N/A |
| Isolde | Croll, 2018 | N/A |
| Locscale | Jakobi et al, 2017 | N/A |
| Phenix | Liebschner et al, 2019 | N/A |
| RELION | Scheres, 2012 | N/A |
| WARP | Tegunov and Cramer, 2019 | N/A |
| **Other** | | |
| EMS100x glow-discharge | Electron Microscopy Sciences | N/A |
| Filter paper | Ted Pella | 47000-715 |
| K3 BioQuantum | Gatan | 1967 |
| Krios G3i | Thermo Fisher Scientific | N/A |
| Vitrobot Mk IV | Thermo Fisher Scientific | N/A |

## Cloning and protein expression

The expression constructs for TRMT10C/SDR5C1 and ELAC2 were as previously described (Reinhard et al, 2017). The various ELAC2 mutations were introduced by site-directed mutagenesis (using Phusion High-Fidelity DNA Polymerase (Thermo Fisher Scientific)). Expression was performed in *E. coli* KRX cells (Promega) in phosphate-buffered Terrific Broth (Sigma-Aldrich) supplemented with 50 μg/ml kanamycin. Expression was induced at $OD_{600}$ 0.8 with 0.5 mM isopropyl β-D-1-thiogalactopyranoside (IPTG) and 0.1% rhamnose overnight at 18 °C.

## Protein purification

For TRMT10C/SDR5C1, cells were lysed in buffer A (50 mM HEPES pH 7.5, 300 mM NaCl, 1 mM β-mercaptoethanol, 5% glycerol, 1 mM IPTG, DNase I (Grade I, Roche)). $Ni^{2+}$ IMAC chromatography (Cytiva) was run with buffer A and eluted with buffer B (20 mM HEPES pH 7.5, 150 mM NaCl, 1 mM β-mercaptoethanol, 5% glycerol, 300 mM imidazole). Next, the sample was directly applied and run through a HiTrapQ column (Cytiva) pre-equilibrated with buffer C (20 mM HEPES pH 7.5, 150 mM NaCl, 5% glycerol, 1 mM DTT) to remove contaminating nucleic acids. The sample was applied on a HiTrap HEPARIN (Cytiva) pre-equilibrated with buffer C and eluted with a 1 M NaCl gradient. The sample was dialyzed overnight in buffer C at 4 °C in the presence of TEV protease (in-house production) and run on a Capto S column (Cytiva) eluted with a 1 M NaCl gradient. Finally, TRMT10C/SDR5C1 was run through a Sephacryl S-300 (Cytiva) in buffer C. TRMT10C/SDR5C1 (with an $OD^{260/280}$ of 0.6) was frozen in liquid nitrogen and stored for later use.

ELAC2 wild-type and mutants were lysed and purified using $Ni^{2+}$ IMAC chromatography as above. Next, ELAC2 was dialyzed in buffer C in the presence of TEV protease. ELAC2 was run through a HiTrapQ column (Cytiva) in buffer C to remove contaminating nucleic acids and purified over a HiTrap Heparin column in buffer C with a salt gradient up to 1 M NaCl. ELAC2 was frozen in liquid nitrogen and stored for later use.

## RNA production

tRNA precursors were prepared by T7 in-vitro transcription. DNA templates were synthesized (Integrated DNA technologies), amplified by PCR (Phusion High-Fidelity DNA polymerase kit, Thermo Fisher Scientific) and purified by phenol:chloroform extraction. Run-off transcription reactions were assembled with in-house purified T7 RNA polymerase (40 mM Tris pH 8, 20 mM $MgCl_2$, 2 mM spermidine, 0.05% tween-20, 10 mM TCEP, 4 mM each nucleotide, 5 μg DNA template) and run for 4 h at 37 °C. The transcripts were purified in a Mono Q ion exchange column (Cytiva) in the buffer 20 mM HEPES pH 7.5, 1 mM $MgCl_2$ and 100 mM NaCl with a gradient up to 1 M NaCl.

## RNase Z cleavage assays

The assays were performed as previously described (Reinhard et al, 2017). Briefly, the reaction was assembled in a buffer containing 20 mM HEPES/KOH pH 7.6, 130 mM KCl, 2 mM $MgCl_2$, 2 mM

TCEP, 5 μM SAM, and 0.1 mg/ml bovine serum albumin (BSA). The RNA substrates (400 nM) were pre-incubated for 10 min at room temperature with TRMT10C/SDR5C1 (1600 nM). Then, they were mixed 1:1 with ELAC2 diluted in the same assay buffer, resulting in the ELAC2 concentrations indicated in each figure. The reactions were incubated for 20 min at room temperature. Proteinase K (RNase-free grade, Thermo Fisher Scientific) was added to each reaction to a final concentration of 0.4 mg/ml and further incubated at 50 °C for 10 min. For the time series assays, the reactions were set up and then aliquots were taken at the indicated times after ELAC2 addition. The reactions were stopped by snap freezing in liquid nitrogen. The samples were then directly incubated at 95 degrees for 5 min to denature ELAC2/SDR5C1/TRMT10C before proteinase K treatment as above. The reactions were analyzed using 15% TBE-UREA PAGE (Thermo Fisher Scientific) and stained with SYBR Green II RNA stain (Thermo Fisher Scientific).

## Mitochondrial RNase Z complex reconstitution

TRMT10C/SDR5C1 was run over an S200 10/300 column (Cytiva) in RNase-free buffer D (20 mM HEPES 7.5, 50 mM NaCl, 1 mM MgCl₂, 1 mM DTT). ELAC2 was purified similarly but in 150 mM NaCl. The complex was assembled at 4 °C with 10 μM TRMT10C/SDR5C1, 20 μM ELAC2, and 20 μM mt-tRNA precursor in buffer D with 200 μM S-(5′-Adenosyl)-L-methionine (SAM) (Sigma-Aldrich). The RNA and ELAC2 preparations contained a higher NaCl concentration; therefore, the salt concentration was adjusted to 50 mM NaCl using the same buffer without NaCl. The complex was applied over the same S200 size exclusion chromatography in buffer D. RNase Z was concentrated by ultrafiltration, supplemented with 0.005% Tween-20 and 200 μM SAM, and directly used for grid preparation.

## Grid preparation and data collection

UltrAuFoil 300 mesh (Quantifoil Micro Tools GMBH; R 1.2/1.3 geometry) grids were glow-discharged at 25 mA for 30 s using an EMS100X glow-discharge unit. Four microliters of the sample with an OD₂₆₀ of 7.5 were applied to the grids and vitrified at 4 °C and 100% humidity using a Vitrobot Mk IV (Thermo Fisher Scientific) (blot 6 s, blot force 3, 595 filter paper (Ted Pella Inc.)).

All cryo-EM data collection was performed with EPU (Thermo Fisher Scientific) using a Krios G3i transmission electron microscope (Thermo Fisher Scientific) operated at 300 kV in the Karolinska Institutet's 3D-EM facility. Images were acquired in nanoprobe 165kX EF-TEM SA mode (0.505 Å/px) with a slit width of 10 eV using a K3 BioQuantum for 1.5 s with 60 fractions and a total fluency of 48 e⁻/Å². Motion correction, dose weighting, CTF estimation, Fourier cropping (to 1.01 Å/px), particle picking (size threshold 100 Å, using the pre-trained BoxNet2Mask_20180918 model), and extraction in 416-pixel boxes were performed on the fly using Warp (Tegunov and Cramer, 2019). Only particles from micrographs with an estimated resolution of 4 Å or better and underfocus between 0.2 and 3 μm were retained for further processing.

## Data processing and model building

The particles were further processed using CryoSPARC v4.1 (Punjani et al, 2017). The RNase Z-HS particles were subjected to 2D classification. Particles from good and bad 2D classes were used to generate one and four ab initio volumes, respectively. The RNase Z-HS and RNase Z^H548A-HS datasets were processed similarly. First, the particle sets were cleaned by heterogeneous refinement using the five ab initio volumes from above. Then, 3D-variability analysis (Punjani and Fleet, 2021) was performed to identify the ELAC2-containing particles. In the case of RNase Z^H548A-HS, two rounds of heterogeneous refinement were performed, and the 3D-variability analysis was replaced by a 3D classification, which fulfilled a similar function in selecting the ELAC2-containing particles. Then, a local refinement focused on ELAC2 was performed. Finally, a 3D classification with a resolution cutoff of 3 Å was performed for the final dataset cleanup. One of the classes yielded a reconstruction with a resolution ~3 Å, while the other classes remained in the range of 4–5 Å and were rejected. The selected particle sets were subjected to a non-uniform refinement with a mask covering the entire RNase Z.

The ELAC2-focused local refinements and the non-uniform refinements were combined in PHENIX (Liebschner et al, 2019). To ensure that ELAC2 is correctly positioned with respect to the rest of the particle, the ELAC2-focused map was fitted into the ELAC2 map from the non-uniform refinement before combining the global and focused maps.

For the RNase Z-HCCA structure, the dataset was processed as follows. First, the particle set was cleaned by heterogeneous refinement in CryoSPARC. The good particles were then imported in RELION-5.0 beta (Scheres, 2012) for 3D auto-refinement and CTF refinement (beamtilt, trefoil, anisotropic magnification and per-particle defocus) using a focus mask on TRMT10C/SDR5C1/mt-tRNA, and Bayesian polishing. The ELAC2-containing particles were then selected by 3D classification using a focus mask on the ELAC2 region in CryoSPARC. These particles were subsequently reimported to RELION-5.0 beta (Scheres, 2012). At this point, a volume was reconstructed and post-processed using the TRMT10C/SDR5C1/mt-tRNA focus mask to obtain the highest resolution possible in the SAM binding site. The resulting map is deposited in EMDB-51230. In parallel, particle subtraction was performed to focus on ELAC2. Then, a 3D classification was performed with local angular searches and a regularization T of 20 to select the ELAC2 particles that refine to high resolution. A final round of per-particle defocus refinement and 3D auto-refine with Blush regularization (Kimanius et al, 2024) was performed. To improve the density of the 3′-CCA trailer, a series of 3D classification runs without changing poses was performed using a focus mask around the active site. At this point, we noticed a mixture of states arising from different truncations of the 3′-CCA trailer, probably arising from impurities in the tRNA preparation. To improve the 3D classification when using a very small mask, one round of supervised 3D classification was performed. One final round of unsupervised 3D classification was performed to ensure no model bias was present in the final ELAC2 reconstruction. The consensus map and the ELAC2-focused map were combined in PHENIX as above, and post-processed using EMReady (He et al, 2023).

For atomic model building, a starting model for RNase Z was generated using the RNase P structure ((Bhatta et al, 2021); PDB 7ONU) for TRMT10C, SDR5C1, and the mt-tRNA precursor, and AlphaFold (Jumper et al, 2021) for ELAC2. The models were built and refined using Coot (Emsley et al, 2010) and PHENIX (Liebschner et al, 2019) against the combined maps. For the low-resolution areas (the ELAC2 exosite), the model was refined using Isolde (Croll, 2018) instead.

The combined maps for RNase Z-HS and RNase Z^H548A-HS were improved iteratively by post-processing with Locscale2 (Jakobi et al, 2017) and model building in Coot/Isolde. Figures were prepared using

ChimeraX, PoseEdit, and ProteinsPlus (Pettersen et al, 2021; Diedrich et al, 2023; Schöning-Stierand et al, 2022).

## Data availability

Cryo-EM reconstruction of RNase Z-HS (composite map): EMDB-50050. Cryo-EM reconstruction of RNase Z-HS (consensus map): EMDB 19954. Cryo-EM reconstruction of RNase Z-HS (ELAC2 focus): EMDB 19955. Atomic model of RNase Z-HS: PDB 9EY0. Cryo-EM reconstruction of RNase Z$^{H548A}$-HS (composite map): EMDB-50051. Cryo-EM reconstruction of RNase Z$^{H548A}$-HS (consensus map): EMDB 19956. Cryo-EM reconstruction of RNase Z$^{H548A}$-HS (ELAC2 focus): EMDB 19957. Atomic model of RNase Z$^{H548A}$-HS: PDB 9EY1. Cryo-EM reconstruction of RNase Z-HCCA (composite map): EMDB-50052. Cryo-EM reconstruction of RNase Z-HCCA (consensus map): EMDB 19958. Cryo-EM reconstruction of RNase Z-HCCA (ELAC2 focus): EMDB 19959. Atomic model of RNase Z-HCCA: PDB 9EY2. Cryo-EM reconstruction of RNase Z-HCCA (TRMT10C/SDR5C1 focus): EMDB-51230. Atomic model of TRMT10C/SDR5C1/mt-tRNA$^{His}$-CCA: PDB 9GCH. Publicly available datasets from Protein Data Bank (7ONU, 4GCW, 1Y44) were used for atomic model building and comparison.

The source data of this paper are collected in the following database record: biostudies:S-SCDT-10_1038-S44318-024-00297-w.

## Peer review information

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

## Acknowledgements

The work was funded by the Knut & Alice Wallenberg Foundation (KAW 2017.0080 and 2018.0080 awarded to BMH) and the Swedish Research Council (VR 2018-03808 and 2022-02326 awarded to BMH). All cryo-EM data was collected at the Karolinska Institutet's 3D-EM facility, and the authors thank A Bondy for excellent support.

## Author contributions

**Genís Valentín Gesé**: Conceptualization; Data curation; Formal analysis; Validation; Investigation; Visualization; Methodology; Writing—original draft; Writing—review and editing. **B Martin Hällberg**: Conceptualization; Data curation; Supervision; Funding acquisition; Validation; Investigation; Writing—original draft; Project administration; Writing—review and editing.

Source data underlying figure panels in this paper may have individual authorship assigned. Where available, figure panel/source data authorship is listed in the following database record: biostudies:S-SCDT-10_1038-S44318-024-00297-w.

## Funding

## Disclosure and competing interests statement

The authors declare no competing interests.

# Expanded View Figures

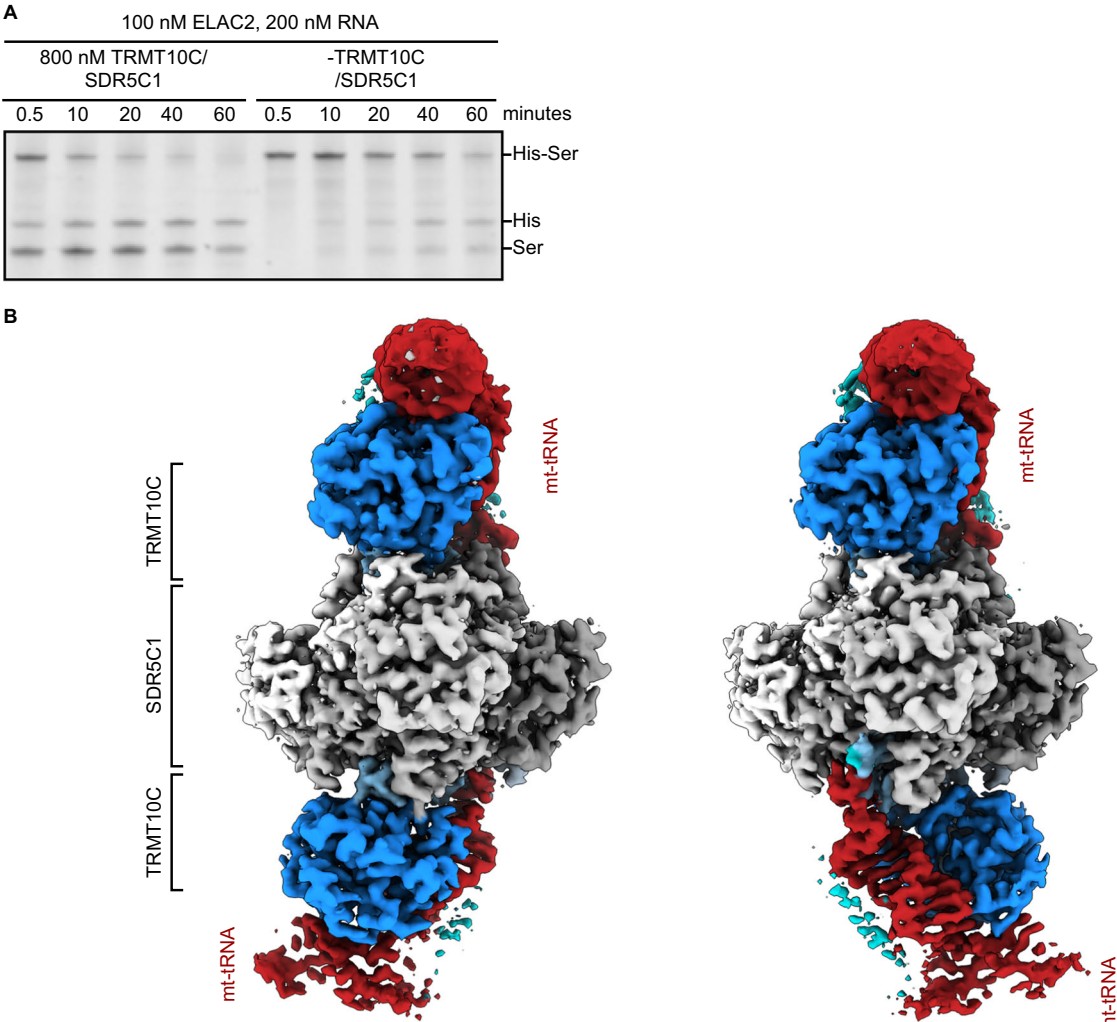

**Figure EV1.  The TRMT10C structure (related to Fig. 1).**

(A) ELAC2 activity in the presence and absence of the TRMT10C/SDR5C1 platform. (B) Cryo-EM density of particles containing two TRMT10C subunits. The 4xSDR5C1 platform offers two mt-tRNA/TRMT10C binding sites. The platform has a vertical and a horizontal twofold symmetry axis. Consequently, it supports two mt-tRNA/TRMT10C subcomplexes, which can be in two different but equivalent orientations (left and right panels). The 4xSDR5C1 are colored in different shades of gray; TRMT10C is colored in dark blue with the NTD highlighted in light blue; the mt-tRNA is in red. ELAC2 density is not visible due to the low occupancy. Source data are available online for this figure.

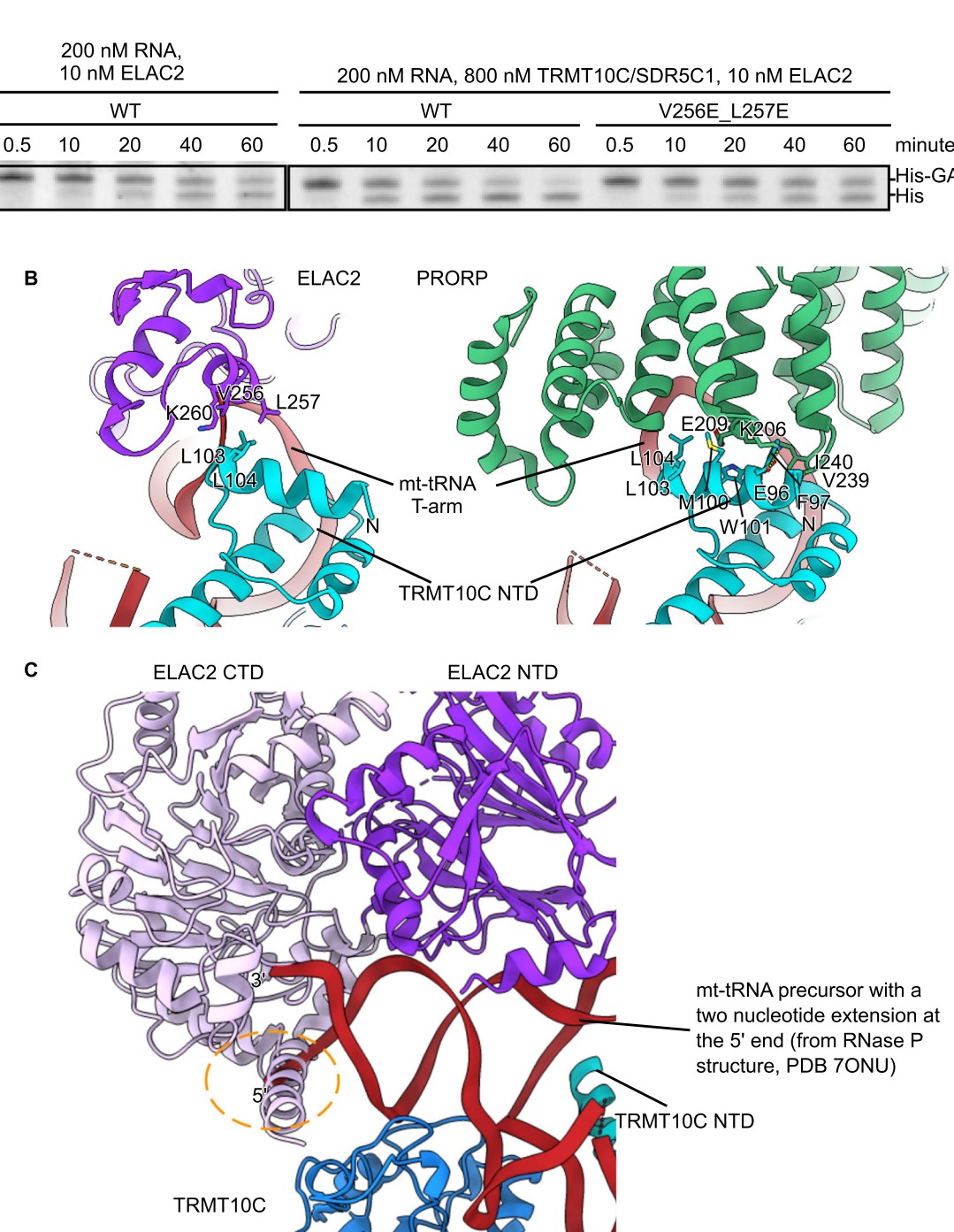

**Figure EV2. TRMT10C/SDR5C1 interactions with ELAC2 and PRORP (related to Fig. 2).**

(A) ELAC2 cleavage of the mt-tRNA$^{His}$-GAG precursor in the absence of TRMT10C/SDR5C1, or with the mutations V256E_L257E that affect residues interacting with TRMT10C. (B) TRMT10C NTD interacts with ELAC2 (RNase Z, left panel) and PRORP (RNase P, right panel) in the T loop region. (C) The ELAC2 C-terminal helix would collide with a 5′ extension on the mt-tRNA precursor (yellow circle). To make this figure, the mt-tRNA precursor in RNase Z was replaced with the RNase P one (mt-tRNA$^{Tyr}$), which has a 2-nucleotide 5′-extension. For this, The RNase Z structure was superimposed on the RNase P structure (Bhatta et al, 2021; PDB 7ONU), using TRMT10C as a reference. Source data are available online for this figure.

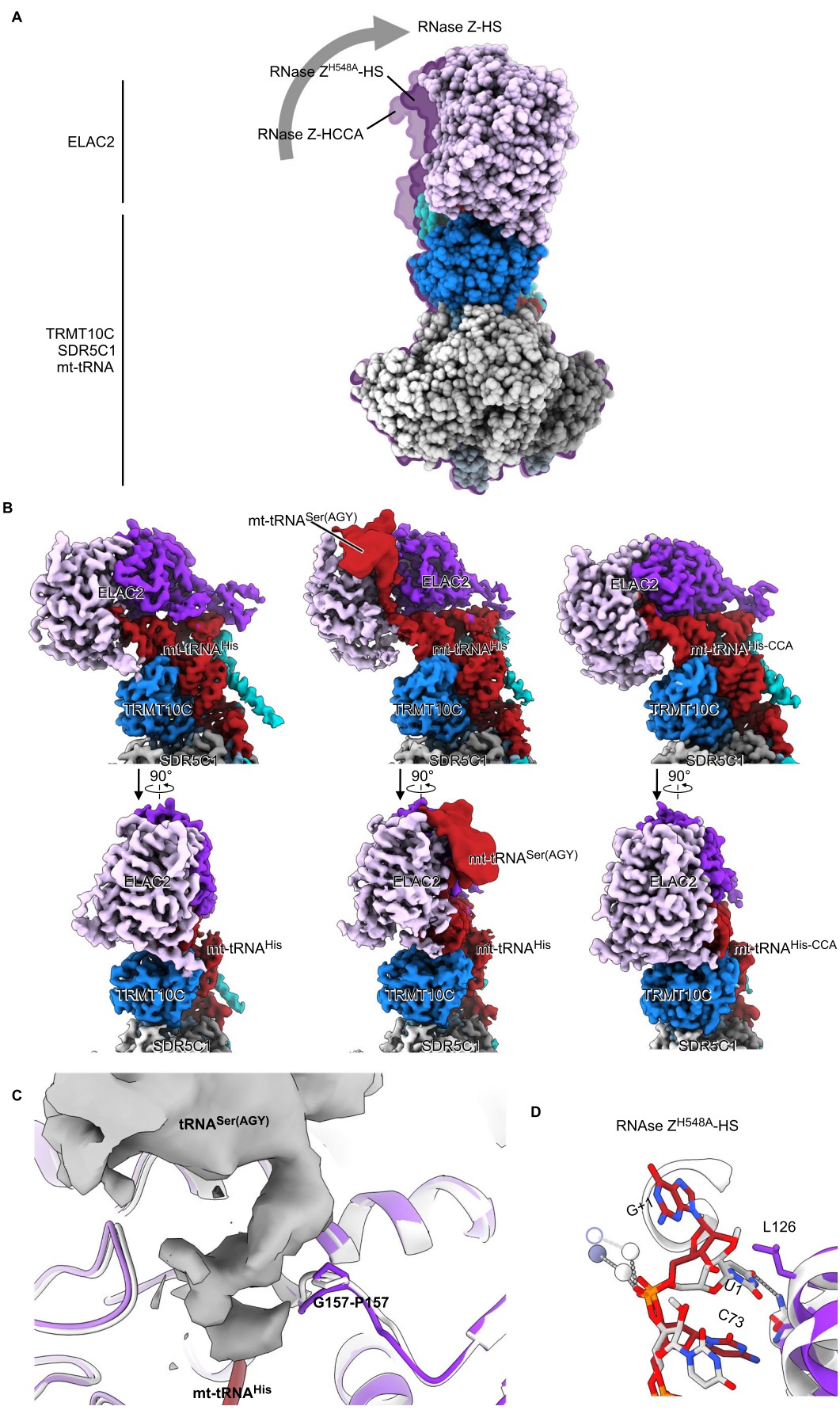

**Figure EV3.  Pivoting of ELAC2 on the mt-tRNA (related to Fig. 3).**

(A) ELAC2 pivots on the mt-tRNA to reach the catalytic position. The RNase Z-HS structure is at the forefront. The outline of the RNase Z$^{H548A}$-HS and the RNase Z-HCCA structures are shown to highlight the movement of ELAC2. ELAC2 is colored in light and dark purple for the CTD and NTD, respectively, while TRMT10C is in blue, and the four SDR5C1 subunits are in shades of gray. (B) Cryo-EM composite maps of RNase Z bound to the three different tRNA precursors, aligned on TRMT10C. The cryo-EM density of the mt-tRNA$^{Ser(AGY)}$ moiety has been Gaussian-filtered in UCSF ChimeraX with a B-factor of 600 for better visualization. To visualize the ELAC2 movement, notice the changes in the gap with TRTM10C. (C) Cryo-EM density for the mt-tRNA$^{Ser(AGY)}$ in RNase Z$^{H548A}$-HS structure. The density, for which it was not possible to create an atomic model, displaces the G157-P157 loop compared with the RNase Z-HS structure (superposed in white). (D) Comparison with the *B. subtilis* RNase Z bound to precursor tRNA. In *B. Subtilis* (white), U1 inserts into a pocket and helps position the scissile bond. In RNase Z$^{H548A}$-HS, G1 does not bind in the pocket as it is partially occluded by L126. The *B. Subtilis* Zn$^{2+}$ are shown as white spheres. The RNase Z$^{H548A}$-HS missing Zn$^{2+}$ is drawn based on the RNase Z-HS structure.

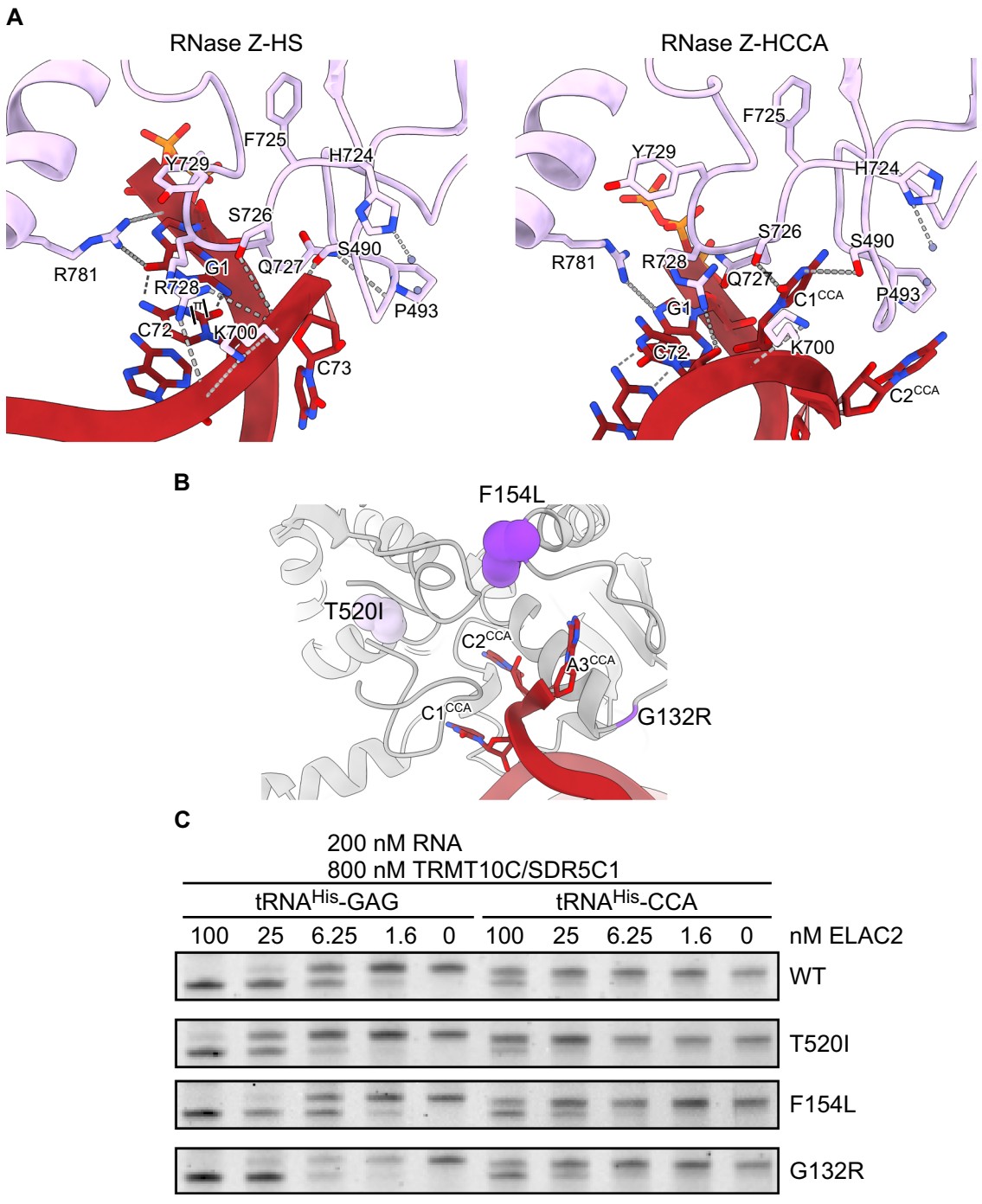

**Figure EV4. Selected ELAC2 mutations involved in hypertrophic cardiomyopathy do not impair the 3′-CCA antidetermination (related to Fig. 5).**

(A) Interactions around the tRNA 3′-end or the 3′-CCA tail involve the same ELAC2 residues. Two parallel black lines indicate π-stacking. Selected electrostatic interactions are indicated with gray dotted lines. (B) Selected clinical mutations near the ELAC2 active site that only mildly impact the catalytic activity, namely T520I (Haack et al, 2013), F154L (Haack et al, 2013), and G132R (Paucar et al, 2018). Mutations are shown in dark and light purple for the NTD and the CTD of ELAC2, respectively. The rest of ELAC2 is in gray, and the tRNA is in red. (C) Cleavage activity of the ELAC2 mutants on tRNA$^{His}$-CCA assessed by TBE-UREA PAGE. None of the mutations significantly affects the 3′-CCA antideterminant effect. The WT panel is the same as in Fig. 5A, shown here for an easier side-to-side comparison. Representative gel images of technical triplicates and biological duplicates for WT. Source data are available online for this figure.

