## [Peer Review File · The EMBO Journal]

Structural basis of 3'-tRNA maturation by the human mitochondrial RNase Z complex

Genis Valentin Gese and B. Martin Hallberg

Corresponding author(s): B. Martin Hallberg (Martin.Hallberg@ki.se)

Review Timeline:

Submission Date:	17th Apr 24
Editorial Decision:	29th May 24
Revision Received:	5th Sep 24
Editorial Decision:	10th Oct 24
Revision Received:	11th Oct 24
Accepted:	16th Oct 24

Editor: *Cornelius Schneider*

Transaction Report:

Dear Dr. Hallberg,

Thank you for submitting your manuscript for consideration by the EMBO Journal. It has now been seen by three referees whose comments are shown below.

All referees agree that the results presented here are of interest and high technical quality while also giving several suggestions that would improve the presentation of the experimental data and make sure that the literature is appropriately discussed. The referees also ask for additional experiments to strengthen the conclusions regarding the cleavage activity and kinetics. I find the assessment by the referees fair and constructive and I would therefore like to invite you to submit a revised version of the manuscript, addressing the comments of all three reviewers. I should add that it is EMBO Journal policy to allow only a single round of revision, and acceptance of your manuscript will therefore depend on the completeness of your responses in this revised version.

Thank you for the opportunity to consider your work for publication. I look forward to your revision.

Yours sincerely,

Cornelius Schneider

Cornelius Schneider, PhD
Editor
The EMBO Journal
c.schneider@embojournal.org

We realize that it is difficult to revise to a specific deadline. In the interest of protecting the conceptual advance provided by the work, we recommend a revision within 3 months (27th Aug 2024). Please discuss the revision progress ahead of this time with the editor if you require more time to complete the revisions. Use the link below to submit your revision:

Referee #1:

Overall Assessment

This manuscript presents a well-written and informative study on the structure and function of human mitochondrial RNase Z in complex with its tRNA substrate and the TRMT10C/SDR5C1 tRNA maturation platform. The authors employ cryo-EM to determine high-resolution structures of the RNase Z complex in different states, revealing the molecular details of tRNA recognition, catalysis, and the antideterminant effect of the 3'-CCA tail.

Unfortunately, coordinates and maps were not provided for review. To gain further insight, I predicted the human mtRNase Z complex with HS, H-w-CCA, H-w/o CCA, and H-w-5'CCCCC using the online AlphaFold 3 server.

Major Concerns Essential to Address

1. Inefficient 3'-CCA Tail Removal by ELAC2 (Line 293): The data suggests that ELAC2 removes the 3'-CCA tail of mt-tRNA^{His-Ser}(AGY) less efficiently compared to its cleavage (Line 293). The figures appear overexposed, making it difficult to identify the product of mt-tRNA^{His} modified with His-CCA. Additionally, as ELAC2 concentration increases, the overall intensity of both the HS and product RNA seems to decrease dramatically compared to the input HS RNA. To address these concerns, please provide:

- A more detailed explanation for these observations, particularly regarding the potential reasons for the decrease in intensity.
- The raw image of panel A from Figure 5 for a more critical evaluation of the data.

2. Discrepancy in Methyltransferase Substrate (Line 138): The authors claim to observe SAH in the methyltransferase active site (Supplementary Fig. 4), while Supplementary Figure 5 shows SAM (not sFigure 4). Since SAM is the typical methyl group donor, it's unclear where the methyl group goes. Does it transfer to the purine-9 position of mt-tRNA^{His}? Please correctly cite the relevant supplementary figure and provide the density map around the ligand to definitively identify the molecule.

3. Structural Support for 5'-to-3' tRNA Processing Order (Abstract): The authors claim that the structural insights provide a molecular basis for the 5'-to-3' tRNA processing order in mitochondria. However, the current evidence seems insufficient. While the data suggests RNase Z clashes with the 5' mt-tRNA, the AlphaFold 3 predicted mt-RNase Z complex with HS and the CCCC-HS RNA suggests a channel on ELAC2 that could potentially accommodate a single-stranded RNA, possibly for 3'-end processing. Please provide more experimental data or detailed reasoning to support the conclusion about the tRNA processing order.

Minor Concern

- Supplementary Figure 1B: The gel electrophoresis method used is likely TBE-UREA PAGE, not TBE-UREA SDS-PAGE.

Referee #2:

In this manuscript, the authors aim to structurally and functionally characterize the human mitochondrial RNase Z complex. Precursor tRNAs are transcribed with leader and trailer sequences, which need to be cleaved by the RNase P complex and RNase Z during the tRNA processing/maturation. In Mitochondria, two large precursor RNA transcripts are transcribed that basically contain all required coding sequences, rRNAs, and tRNAs. The polycistronic transcripts are cleaved by the mitochondrial RNase P and RNase Z complexes to generate the individual coding and non-coding RNA entities. The structure

of the human mitochondrial RNase P was presented a few years ago, which showed the arrangement of the different subunits, namely TRMT10C, SDR5C1 and PRORP. After tRNA 5'-processing by RNase P, RNase Z processes the 3'-trailer. Human RNase Z is produced in two forms (ELAC1 and ELAC2), but only ELAC2 can enter mitochondria. The authors have previously shown that most human mitochondrial tRNA precursors remain bound to TRMT10C/SDR5C1 after the cleavage by RNase P (PROP). This association enhances the processing activity of ELAC2 for most mitochondrial tRNAs.

In the presented manuscript, the authors determine the single particle cryo-EM structure of the mitochondrial RNase Z after cleavage of the mt-tRNA precursor (tRNA^{His}-tRNA^{Ser}(AGY)) - representing the product state. They analyse the observed structural assembly that contains the TRMT10C-SDR5C1 subcomplex, mt-tRNA^{His} and ELAC2 in great detail. In addition, they present the structure of a RNase Z complex mutant in complex with the mt-tRNA^{His}-tRNA^{Ser}(AGY) precursor. Last but not least, they present a very high-resolution cryo-EM structure (2.1 Å) of RNase Z in complex with mt-tRNA^{His} after addition cleavage and addition of the 3' -CCA tail. They compare these structures to understand the role of residues in ELAC2 that coordinate the cleavage and how the complex avoids an endless cycling between RNase Z cleavage and addition of CCA by TRNT1. The structural work is elegantly complemented by activity assays.

Recent work by the Tisne and Hillen labs presented similar structures in preprints and after getting myself familiar with all three studies, I conclude that the presented work very nicely complements the work of the other teams. The mutant structure with the HS-precursor seems to be almost identical to another structure presented in one of the other manuscripts - however, the mutants appears to be different and the focus of the three manuscripts is highly synergistic and not redundant. I highly appreciate the detailed discussion that highlights the other studies and compares the results in a transparent and reasonable way. In summary, the presented work by Gesé and Hällberg represents an important molecular milestone in understanding tRNA processing in our mitochondria and I would suggest publication of the work in EMBO Journal after my comments are addressed and resolved.

Major

1. I am not fully convinced about the functional analyses for the 3'-CCA discrimination and the interpretation by the authors. Their data as such is clear, but does not exclude different other mechanisms that could break the cycle between RNase Z cleavage and CCA addition. For instance, TRNT1 could play a similar protective function after the addition of the CCA and binding of TRNT1 could simply limit the rebinding of ELAC2. It would be interesting, whether other sequences (e.g. a scrambled CCA sequence, CCU (mentioned in the discussion) or any other 3-nucleotide stretch), would have a similar protective function or whether the CCA tail is exclusively displaced and protected from cleavage.
2. Did the authors test the cleavage activity of the clinical mutants that are mentioned on page 8. Some of this mutants might also be relevant for the CCA 3'-CCA discrimination and I was wondering, both HS as well as HCCA could be tested with these mutants.

Minor

- The RNase Z-HS mutant structure is almost absent from the manuscript and I think the authors should show an overall structure that show whether they can see the tRNA^{Ser}(AGY) part bound in ELAC2.
- Similar to the above comment, I think it would be nice to show an overall view of the RNase Z-HCCA structure. I would even suggest to show the three structures side by side to make it easier for non-structural biologists to recognize that three different structures have been determined - this is currently a bit lost in the manuscript.
- I am not a big fan of frankenstein/composite maps as they do not reflect the actual data. In understand that the quality of the final maps looks better and the authors do not hide their processing pipelines, but I would like to authors to check available literature on the recommendation against using this kind of maps and reconsider this approach for their future work.
- It is not clear for me how the HCCA structure can have a resolution of 2.1 Å and a "map resolution range" between 2.2 and 7 Å?
- I guess the defocus range should be negative?

Referee #3:

The manuscript by Gesé and Hällberg presents original structural insight into the function of the long form of human RNase Z encoded by the gene ELAC2. It is the first structure of a long form of RNase Z with a bound pre-tRNA substrate and thereby allows significant insights into the functioning of the enzyme. This is nicely illustrated by the insights into the CCA-antideterminant effect, or the potential clashes of the C-terminal α helix with a 5' leader as well as interactions with the 5'-phosphate, which together explain why 5'-end processing of pre-tRNAs has to precede 3'-end processing.

Issues/concerns that should be addressed

References are either cited by "author, year", when referring to references found in the manuscript, or by their DOI, if not found in the manuscript.

- The authors build their manuscript and its conclusions to a large extent on a concept they had introduced in 2017 (Reinhard et al., 2017) suggesting that TRMT10C-SDR5C1 acts as a "tRNA-maturation platform" in human mitochondria, where the 3 main

processing proteins (PRORP, ELAC2 and TRNT1) interact sequentially with TRMT10C/SDR5C1 and/or the pre-tRNA bound to it, to constitute mtRNase P, mtRNase Z and the mitochondrial CCA-adding enzyme. While this model appears attractive at first sight, it is exclusively built on the finding that the RNase Z activity of ELAC2 on some pre-tRNAs appears to be stimulated by TRMT10C/SDR5C1 and that both ELAC2 and TRNT1 are able to bind to a TRMT10C-SDR5C1-pre-tRNA complex, although this appears to be of no functional relevance in the case of CCA addition (Reinhard et al., 2017). While the structure presented here now confirms that ELAC2 can indeed bind to a TRMT10C/SDR5C1-pre-tRNA complex, it does not reveal what the functional relevance of this interaction is. The authors apparently deliberately ignore that in 2023 conflicting data have been published (10.1093/nar/gkad713) that question this model. It would be appropriate to mention and discuss these results in appropriate places (e.g., lines 74-84 and other occasions). Furthermore, ELAC2 (and TRNT1) also act on nuclear encoded pre-tRNAs in vivo, obviously in the absence of TRMT10C/SDR5C1. Although this is briefly mentioned, it is not taken into account in the manuscript's conclusions and discussion (see, e.g., lines 329-344). Thereby the mechanistic role of the TRMT10C/SDR5C1 complex for processing of mt-tRNAs by ELAC2 remains mostly unclear. All the enzymatic data previously published by the authors (Reinhard et al., 2017) and the few included in this manuscript are single-endpoint assays using extremely high enzyme and substrate concentrations. At least some basic kinetic analyses, in line with those experiments recently published for the processing by PRORP and its dependency on TRMT10C/SDR5C1 (10.1093/nar/gkad713), should be included to substantiate the mechanistic point. In the light of conflicting data and the absence of further evidence, the functional/mechanistic significance of the coincident binding of ELAC2 to the TRMT10C-SDR5C1-pre-tRNA complex, while obviously beneficial for determining the cryo-EM structure of the enlarged ELAC2 complex, remains unclear.

- "RNase" is mis-capitalized in many places as "RNase" including the title
- Canonical numbering of tRNA nucleotide positions should be used in text and figures; e.g., the discriminator base is number 73 by general convention and not 69. The same applies to the numbering of the T loop nucleotides (should be 54-60), etc.
- The interactions of TRMT10C with PRORP and ELAC2 should be compared. Such comparison would warrant a paragraph in the discussion and could be potentially insightful.
- Line 35: "tRNA punctuation model" not "punctate model"
- Line 37: Hällberg & Larsson, 2014 is an inappropriate self-reference here. A newer review comparing the different forms of RNase P would be more appropriate here (e.g., 10.1007/978-3-030-57246-4_11) or individual reviews of the two forms of RNase P (e.g., 10.1016/j.jbc.2024.105729 and 10.1016/j.jbc.2024.105731)
- Line 40: reference (Bartkiewicz et al., 1989) is in the wrong place; move behind the comma in line 39.
- Line 42: add 10.1093/nar/gkad713, where this was finally demonstrated.
- Line 44: reference missing; add 10.1093/nar/gks910 after "co-substrate".
- Lines 46-47: This has recently been shown to be an artifact (see 10.1093/nar/gkad713).
- Line 50: "prokaryotes" is outdated terminology; better use bacteria and archaea.
- Line 52: better "in many eukaryotes" than "in most ...", as it is not yet clear how widespread the small form is in eukaryotes.
- Line 64: incomplete sentence!
- Line 69: delete RNA from "nuclear RNA processing".
- Lines 72-73: ELAC1 is unlikely to compensate for ELAC2 deficiency in the nucleus, but there is another major nuclear tRNA 3'-end processing pathway that has to be mentioned in this context.
- Lines 130-133: I do not see how and why the similarity of the SDR5C1/TRMT10C/tRNA complex in the structures with PRORP and with ELAC2 would support or "highlight" a coupling of the 2 reactions. In fact, I do not see any evidence for a true coupling of the reactions.
- Lines 165-167: What is the Ψ -loop? I suppose it should either read T loop or T Ψ C loop. Anyway, tRNA-His has a canonically sized T loop of 7 nt; only 2 mitochondrial tRNAs have T loop of 8 nt and one of 9 nt. I am not convinced that would lead to base stacking with F254 as suggested.
- Lines 169-174: I am not convinced of the hydrophobic interactions, particularly in case of K260, because of the distances as apparent in the figure. One would require access to the PDB file to evaluate this. Are L103 and L104 in TRMT10C also conserved in metazoans?
- Lines 187-190: Are R788 and R791 in ELAC2, and Q343 in TRMT10C conserved in metazoans?
- Line 193: Add reference (Rossmann et al., 1995) that first showed the processing order.
- Lines 252-258: To support the conclusions the acceptor stem of mt-tRNA-Ser(AGY) should be modeled; the cryo-EM densities are insufficient.
- Lines 290-293 and Figure 5: The design of the experiment does not allow to distinguish between effects of trailer sequence CCA versus GAG (in the case of mt-tRNA-Ser(AGY)) and the different length of trailer. A tRNA-His with a GAG trailer should be used in comparison with the tRNA-His with CCA. That would also show whether the resolution of the gel as presented would at all have been sufficient to see cleavage of the 3 nt from the tRNA-His 3' end. This is a simple experiment that should be included.
- Lines 347-358: It has recently been shown that 5'-end processing of mt-tRNA-Ser(UCN) by PRORP is also stimulated by TRMT10C/SDR5C1 (10.1093/nar/gkad713). Thus, mt-tRNA-Ser(UCN) can obviously also be bound by TRMT10C/SDR5C1. Moreover, mt-tRNA-Gln requires TRMT10C/SDR5C1 for 5'-end processing by PRORP (10.1093/nar/gkad713), which is also at odds with the proposed model.
- Lines 360-364: As pointed out above, kinetic experiments would indeed be crucial to support the model the manuscript is built on. Speculations do not appear to be sufficient here.
- Lines 390-400: Largely redundant with the results. Discussion could be shortened here.

Point-by-point-response to reviewers

Referee #1:

Overall Assessment

This manuscript presents a well-written and informative study on the structure and function of human mitochondrial RNase Z in complex with its tRNA substrate and the TRMT10C/SDR5C1 tRNA maturation platform. The authors employ cryo-EM to determine high-resolution structures of the RNase Z complex in different states, revealing the molecular details of tRNA recognition, catalysis, and the antideterminant effect of the 3'-CCA tail.

Unfortunately, coordinates and maps were not provided for review. To gain further insight, I predicted the human mtRNase Z complex with HS, H-w-CCA, H-w/o CCA, and H-w-5'CCCCC using the online AlphaFold 3 server.

We shared the models and maps through the editor during the review process, and they should be accessible in the review interface. In our hands, AF3 gives mixed results, some models are reasonable, while others are clearly not accurate.

Major Concerns Essential to Address

1. Inefficient 3'-CCA Tail Removal by ELAC2 (Line 293): The data suggests that ELAC2 removes the 3'-CCA tail of mt-tRNA^{His}-Ser(AGY) less efficiently compared to its cleavage (Line 293). The figures appear over posed, making it difficult to identify the product of mt-tRNA^{His} modified with His-CCA. Additionally, as ELAC2 concentration increases, the overall intensity of both the HS and product RNA seems to decrease dramatically compared to the input HS RNA. To address these concerns, please provide:

- A more detailed explanation for these observations, particularly regarding the potential reasons for the decrease in intensity.

- The raw image of panel A from Figure 5 for a more critical evaluation of the data.

We have finetuned this gel analysis during revision, and we believe the results are better now. Following a suggestion from reviewer 3, we are now comparing the removal of the CCA tail to a 3'-GAG tail, as they are comparable in size. The results are in Fig 5A. Furthermore, we have uploaded the raw images with our revised manuscript.

2. Discrepancy in Methyltransferase Substrate (Line 138): The authors claim to observe SAH in the methyltransferase active site (Supplementary Fig. 4), while Supplementary Figure 5 shows SAM (not sFigure 4). Since SAM is the typical methyl group donor, it's

unclear where the methyl group goes. Does it transfer to the purine-9 position of mt-tRNA^{His}? Please correctly cite the relevant supplementary figure and provide the density map around the ligand to definitively identify the molecule.

During grid preparation, the TRMT10C methyltransferase activity will start at the same time as the required components are mixed; therefore, the relative proportions of purine-9 methylation and SAM/SAH will vary with the incubation time before vitrification. Since we focused on the nuclease activity in this work, we didn't specifically aim to obtain one state over the other.

Nevertheless, we sought to investigate the methyltransferase active site in more detail during the revision process. To this end, we focused on the TRMT10C active site to generate a reconstruction at 1.9Å. Inspection of the map suggests a mixture of SAH and SAM. While SAM is strongly predominant, traces of N1 methylation can also be observed. From this, we conclude that the methylation reaction occurred in a subset of particles. The new reconstruction is now included in Table EV1 and deposited with EMDB code 51230, and the associated model with PDB code 9GCH. The cryo-EM density around the SAM/SAH and a Posedit illustration are shown in Appendix Fig S4.

3. Structural Support for 5'-to-3' tRNA Processing Order (Abstract): The authors claim that the structural insights provide a molecular basis for the 5'-to-3' tRNA processing order in mitochondria. However, the current evidence seems insufficient. While the data suggests RNase Z clashes with the 5' mt-tRNA, the AlphaFold 3 predicted mt-RNase Z complex with HS and the CCCC-HS RNA suggests a channel on ELAC2 that could potentially accommodate a single-stranded RNA, possibly for 3'-end processing. Please provide more experimental data or detailed reasoning to support the conclusion about the tRNA processing order.

First of all, AlphaFold3 results can be misleading. There is indeed a channel for the 3'-end processing that we show in the paper (the active site is located in this channel). Regarding the 5'-end, for a +1 nt to fit, the RNA backbone would have to be highly strained to avoid clashing into the C-term of ELAC2. There is no possibility for extending further, i.e. a +2 nt in the tRNA 5'-end, at least not in the TRM10TC/SDR5C1/ELAC2 complex as it would clash into the U66 and A67 that are fixed from the backside by TRMT10C. In short, there is some space when looking at a molecular surface, but when the limitations on available backbone angles for RNA are factored in, the possibilities rapidly diminish, at least for tRNAs bound to the TRM10TC/SDR5C1 platform.

The tRNA processing order is well supported by the existing literature from a biochemical point of view:

Rackham et al., 2016 10.1016/j.celrep.2016.07.031

Siira et al., 2018 10.15252/embr.201846198

Rossmannith W., 2011 10.1371/journal.pone.0019152
Rossmannith et al., 1995 10.1074/jbc.270.21.12885

In addition, recent structural work published by Meynier et al. 2024 (10.1038/s41467-024-49132-0) and in a BioRxiv preprint by Bhatta et al., 2024 (<https://doi.org/10.1101/2024.04.04.588063>) also made the same observation as we did, i.e., that the ELAC2 C-terminal helix would collide with a 5'-precursor.

Minor Concern

- Supplementary Figure 1B: The gel electrophoresis method used is likely TBE-UREA PAGE, not TBE-UREA SDS-PAGE.

We thank the reviewer for this observation and have corrected the text accordingly.

Referee #2:

In this manuscript, the authors aim to structurally and functionally characterize the human mitochondrial RNase Z complex.

Precursor tRNAs are transcribed with leader and trailer sequences, which need to be cleaved by the RNase P complex and RNase Z during the tRNA processing/maturation. In Mitochondria, two large precursor RNA transcripts are transcribed that basically contain all required coding sequences, rRNAs, and tRNAs. The polycistronic transcripts are cleaved by the mitochondrial RNase P and RNase Z complexes to generate the individual coding and non-coding RNA entities. The structure of the human mitochondrial RNase P was presented a few years ago, which showed the arrangement of the different subunits, namely TRMT10C, SDR5C1 and PRORP. After tRNA 5'-processing by RNase P, RNase Z processes the 3'-trailer. Human RNase Z is produced in two forms (ELAC1 and ELAC2), but only ELAC2 can enter mitochondria. The authors have previously shown that most human mitochondrial tRNA precursors remain bound to TRMT10C/SDR5C1 after the cleavage by RNase P (PROP). This association enhances the processing activity of ELAC2 for most mitochondrial tRNAs.

In the presented manuscript, the authors determine the single particle cryo-EM structure of the mitochondrial RNase Z after cleavage of the mt-tRNA precursor (tRNA^{His}-tRNA^{Ser}(AGY)) - representing the product state. They analyse the observed structural assembly that contains the TRMT10C-SDR5C1 subcomplex, mt-tRNA^{His} and ELAC2 in great detail. In addition, they present the structure of a RNase Z complex mutant in complex with the mt-tRNA^{His}-tRNA^{Ser}(AGY) precursor. Last but not least, they present a very high-resolution cryo-EM structure (2.1 Å) of RNase Z in complex with mt-tRNA^{His} after addition cleavage and addition of the 3' -CCA tail. They compare these structures to understand the role of residues in ELAC2 that coordinate the cleavage and

how the complex avoids an endless cycling between RNase Z cleavage and addition of CCA by TRNT1. The structural work is elegantly complemented by activity assays.

Recent work by the Tisne and Hillen labs presented similar structures in preprints and after getting myself familiar with all three studies, I conclude that the presented work very nicely complements the work of the other teams. The mutant structure with the HS-precursor seems to be almost identical to another structure presented in one of the other manuscripts - however, the mutants appears to be different and the focus of the three manuscripts is highly synergistic and not redundant. I highly appreciate the detailed discussion that highlights the other studies and compares the results in a transparent and reasonable way. In summary, the presented work by Gesé and Hällberg represents an important molecular milestone in understanding tRNA processing in our mitochondria and I would suggest publication of the work in EMBO Journal after my comments are addressed and resolved.

Major

1. I am not fully convinced about the functional analyses for the 3'-CCA discrimination and the interpretation by the authors. Their data as such is clear, but does not exclude different other mechanisms that could break the cycle between RNase Z cleavage and CCA addition. For instance, TRNT1 could play a similar protective function after the addition of the CCA and binding of TRNT1 could simply limit the rebinding of ELAC2. Yes, TRNT1 will likely protect the CCA tail in its ternary product complex, but in our previous work (published and unpublished), we have found that the complex with TRNT1 is quite weak (e.g., Reinhard et al., NAR 2017, please compare Figure 4H and 5G). Hence, protection through a TRNT1-product complex is unlikely to have more than a marginal role in the large scheme of tRNA processing in human mitochondria, while the significant difficulty (see Fig 5A) for ELAC2 to process CCA-tailed substrates would have a comparatively large effect.

It would be interesting, whether other sequences (e.g. a scrambled CCA sequence, CCU (mentioned in the discussion) or any other 3-nucleotide stretch), would have a similar protective function or whether the CCA tail is exclusively displaced and protected from cleavage.

Previous extensive analyses by Nashimoto 1997 (10.1093/nar/25.6.1148) and Mohan et al. 1999 (10.1093/nar/25.6.1148) using the pig, mouse, and drosophila RNase Z demonstrated the following: i) the first nucleotide has to be a C has the strongest antideterminant effect; ii) a C or U in the second position enhances the antideterminant effect; iii) the third position does not contribute to the antideterminant effect and can be any nucleotide. We think that these also apply to the human ELAC2, given the high

sequence conservation between humans, mice, and pigs. Accordingly, our structure with the CCA tail shows that i) the first C has full H-bonding, explaining why it has the strongest antideterminant effect, while U is excluded by mismatching H-bonds and A/G are excluded due to their size ii) the second position allows only C/U but excludes A/G due to their size. We have now expanded the discussion to further clarify (lines 344-347).

2. Did the authors test the cleavage activity of the clinical mutants that are mentioned on page 8.

Careful biophysical assays of these clinical mutations have already been published (Saoura et al., 2019). Still, we tested the subset of these mutations that was shown to have a mild effect on the cleavage reaction to answer the question below (results shown in EV4B and C). The motivation for focusing on the mutations with mild effect on the nuclease reaction but with clinical phenotypes was that the clinical mutations with a strong effect on the nuclease activity would likely be hindered in cleaving the CCA-tail as well as the natural substrate sequence, making the assay uninformative.

Some of these mutants might also be relevant for the CCA 3'-CCA discrimination and I was wondering, both HS as well as HCCA could be tested with these mutants.

We have investigated the activity of mutations close to the active site whose enzymatic activity is similar to or only slightly lower than the wild-type (Saoura et al., 2019). For these mutants, an impaired 3'-CCA antidetermination could be the underlying cause of the clinical manifestations. In conclusion, we found no significant effect on the 3'-CCA antidetermination for these mutants. The results are shown in Fig EV4C.

Minor

- The RNase Z-HS mutant structure is almost absent from the manuscript and I think the authors should show an overall structure that shows whether they can see the tRNA^{Ser}(AGY) part bound in ELAC2.

We can see parts of the tRNA^{Ser}(AGY), shown in Fig 4A and Fig EV3B,C. When it comes to the H548A mutant structure, the relative weight we assign to it in our paper is based on the significant changes one can observe when the active site structure is perturbed. An active-site mutant ternary structure is easier to obtain practically than a wt product structure or a CCA-tail ternary structure. However, an active-site mutant ternary structure provides less relevant information on catalysis due to the significant active site perturbation.

- Similar to the above comment, I think it would be nice to show an overall view of the RNase Z-HCCA structure. I would even suggest to show the three structures side by

side to make it easier for non-structural biologists to recognize that three different structures have been determined - this is currently a bit lost in the manuscript.

Thank you for this suggestion. We have now presented an overview of the RNaseZ^{H548A}-HS and RNaseZ-HCCA structures in Fig 4 and 5, respectively. Furthermore, all the three structures are shown side by side in Fig EV3B.

- I am not a big fan of frankenstein/composite maps as they do not reflect the actual data. In understand that the quality of the final maps looks better and the authors do not hide their processing pipelines, but I would like to authors to check available literature on the recommendation against using this kind of maps and reconsider this approach for their future work.

We thank you for this comment, and we agree in principle. We will always deposit overall maps and maps from focused refinement in the EMDB depositions. However, the way the interlink between the PDB and EMDB is set up, there is only one map that comes up for the casual PDB user. If that map is not of reasonable quality in the relevant sections discussed in the accompanying paper, there is a risk for confusion on the relative quality of the underlying reconstruction. We hope that this access mode will be improved in the future and have raised these concerns with personnel at the EBI.

- It is not clear for me how the HCCA structure can have a resolution of 2.1 Å and a "map resolution range" between 2.2 and 7 Å?

We thank the reviewer for this observation, which has now been addressed in an updated table.

- I guess the defocus range should be negative?

It is indeed negative. "Defocus" is now changed to "Underfocus" in Table EV1.

Referee #3:

The manuscript by Gesé and Hällberg presents original structural insight into the function of the long form of human RNase Z encoded by the gene ELAC2. It is the first structure of a long form of RNase Z with a bound pre-tRNA substrate and thereby allows significant insights into the functioning of the enzyme. This is nicely illustrated by the insights into the CCA-antideterminant effect, or the potential clashes of the C-terminal α helix with a 5' leader as well as interactions with the 5'-phopshate, which together explain why 5'-end processing of pre-tRNAs has to precede 3'-end processing.

Issues/concerns that should be addressed

References are either cited by "author, year", when referring to references found in the manuscript, or by their DOI, if not found in the manuscript.

- The authors build their manuscript and its conclusions to a large extent on a concept they had introduced in 2017 (Reinhard et al., 2017) suggesting that TRMT10C-SDR5C1 acts as a "tRNA-maturation platform" in human mitochondria, where the 3 main processing proteins (PRORP, ELAC2 and TRNT1) interact sequentially with TRMT10C/SDR5C1 and/or the pre-tRNA bound to it, to constitute mtRNase P, mtRNase Z and the mitochondrial CCA-adding enzyme. While this model appears attractive at first sight, it is exclusively built on the finding that the RNase Z activity of ELAC2 on some pre-tRNAs appears to be stimulated by TRMT10C/SDR5C1 and that both ELAC2 and TRNT1 are able to bind to a TRMT10C-SDR5C1-pre-tRNA complex, although this appears to be of no functional relevance in the case of CCA addition (Reinhard et al., 2017).

In our previous work (Reinhard et al., 2017), we found a positive effect of the TRMT10C-SDR5C1 tRNA maturation platform on the ELAC2 for 17 mitochondrial tRNA substrates and a similar assay reported by the Hillen group confirmed these results recently (10.1101/2024.04.04.588063). However, the effects of the platform on CCA-addition by TRNT1 were only studied for two substrates in Reinhard et al., 2017. Back then, we could only find a positive effect in terms of specificity (not activity broadly defined) of the TRMT10C/SDR5C1 platform at non-physiological ionic strength.

Since the effects of the TRMT10C-SDR5C1 platform have only been assayed for two substrates, it is, in our opinion, rather premature to state that there is no functional relevance for the platform in terms of CCA-addition for *all* human mitochondrial tRNA substrates. We note that during the revision of this manuscript, a low-resolution reconstruction of TRNT1 bound to mt-tRNA-Ile and TRMT10C/SDR5C1 has been published (Meynier et al., 2024). It is not mentioned in that publication why specifically mt-tRNA-Ile was chosen as a substrate for the cryo-EM work, for example, if any substrate scouting was performed to find a tRNA substrate that mediated stronger binding of TRNT1 to the platform. Nevertheless, TRNT1 is not the focus of the work presented here.

While the structure presented here now confirms that ELAC2 can indeed bind to a TRMT10C/SDR5C1-pre-tRNA complex, it does not reveal what the functional relevance of this interaction is. The authors apparently deliberately ignore that in 2023 conflicting data have been published (10.1093/nar/gkad713) that question this model. It would be appropriate to mention and discuss these results in appropriate places (e.g., lines 74-84 and other occasions).

Thank you for bringing this work by the Rossmann group to our attention. We find it inappropriate to be accused of deliberately ignoring it, especially since ELAC2 was not mentioned in either the title or the abstract in 10.1093/nar/gkad713. We have incorporated the reference where appropriate.

We think that a post-publication review of the recent work by the Rossmann group is definitely outside the scope of our manuscript and, therefore, also this point-by-point response but we can comment within the context of the present work and refer to other preprinted and published work in our response.

We note that the inhibitory effect of TRMT10C/SDR5C1 on ELAC2 activity for tRNA(Ala) shown in 10.1093/nar/gkad713 was not observed in Reinhard et al., 2017 or supported by the recent work by Hillen's group (10.1101/2024.04.04.588063). In contrast, these two groups, independently, showed that the TRMT10C/SDR5C1 complex is stimulatory for ELAC2 activity on tRNA(Ala).

Furthermore, the two time-series assays performed during the revision of this manuscript clearly show that TRMT10C/SDR5C1 significantly enhances ELAC2 catalysis rate, one on the substrate used for the structural work (EV1A) and one with a very short 3'-stretch (EV2A) used for the characterization of the clinically relevant mutants. Furthermore, we tested the importance of the hydrophobic interaction between the TRMT10C NTD and the outside of the ELAC2 exosite by mutating two residues (V256E/L257E) specifically involved in the TRMT10C-ELAC2 hydrophobic interaction but not in T-loop binding.

In addition, we note that two other manuscripts on the structure of the RNase Z complex (Meynier et al., Nat Com 2024 and Bhatta et al., BioRxiv 2024) using active-site mutants strongly confirm the results of Reinhard et al, 2017 and further emphasize the importance of TRMT10C/SDR5C1 as a tRNA maturation platform in human mitochondria.

Lastly, more as a comment at this stage and in this context of the point-by-point response, due to the strong expression-host tRNA contamination of the TRMT10C/SDR5C1 complex if only IMAC purification is used, we recommend adding the OD260/280 ratio of TRMT10C-SDR5C1 complexes used for biochemistry in the corresponding methods sections. The OD260/280 ratio for the TRMT10C/SDR5C1 complex preparations used in this work was 0.6; M&M line 398. Specifically, to obtain reproducible results in biochemical assays, we find it essential to remove expression-host nucleic acids from the TRMT10C/SDR5C1 complex using either heparin or cation-exchange column chromatography.

Furthermore, ELAC2 (and TRNT1) also act on nuclear encoded pre-tRNAs in vivo, obviously in the absence of TRMT10C/SDR5C1.

Yes, but these nuclear-encoded tRNAs are of type 0 and, hence, do not need TRMT10C/SDR5C1 (see discussion).

Although this is briefly mentioned, it is not taken into account in the manuscript's conclusions and discussion (see, e.g., lines 329-344). Thereby the mechanistic role of the TRMT10C/SDR5C1 complex for processing of mt-tRNAs by ELAC2 remains mostly unclear. All the enzymatic data previously published by the authors (Reinhard et al., 2017) and the few included in this manuscript are single-endpoint assays using extremely high enzyme and substrate concentrations. At least some basic kinetic analyses, in line with those experiments recently published for the processing by PRORP and its dependency on TRMT10C/SDR5C1 (10.1093/nar/gkad713), should be included to substantiate the mechanistic point.

During revision, we performed two time-series assays that clearly show that TRMT10C-SDR5C1 significantly enhances ELAC2 catalysis rate, one on the substrate used for the structural work (EV1A) and one with a very short 3'-stretch (EV2A) used for the characterization of the clinically relevant mutants. Furthermore, we tested the importance of the hydrophobic interaction between the TRMT10C NTD and the outside of the ELAC2 exosite by mutating two residues (V256E/L257E) specifically involved in the TRMT10C-ELAC2 hydrophobic interaction but not in T-loop binding.

In the light of conflicting data and the absence of further evidence, the functional/mechanistic significance of the coincident binding of ELAC2 to the TRMT10C-SDR5C1-pre-tRNA complex, while obviously beneficial for determining the cryo-EM structure of the enlarged ELAC2 complex, remains unclear.

We have now performed a time course of tRNA 3'-cleavage by ELAC2 in the presence and absence of TRMT10C/SDR5C. These are shown in Figures EV1A and EV2A. We observe that the ELAC2 activity is higher in the presence of TRMT10C/SDR5C for both the His-Ser(AGY) precursor (EV1A) and the His-GAG precursor (EV2A).

Furthermore, based on the structure, we introduced two mutations (V256E/L257E) on the outside of the exosite domain that do not affect ELAC2 tRNA binding but only the interaction with the TRMT10C in the complex (EV2B). We find that these mutants affect the catalysis rate of ELAC2 in the presence of TRMT10C/SDR5C1 to a level comparable to wild-type ELAC2 alone (EV2A).

- "RNase" is mis-capitalized in many places as "RNase" including the title
Thank you for this observation. We have now implemented consistent capitalization of RNase.

- Canonical numbering of tRNA nucleotide positions should be used in text and figures; e.g., the discriminator base is number 73 by general convention and not 69. The same applies to the numbering of the T loop nucleotides (should be 54-60), etc.
We updated the numbering as suggested. For the non-expert reader, a translation table between the PDB nucleotide numbering and the canonical numbering is in Appendix Table S1.

- The interactions of TRMT10C with PRORP and ELAC2 should be compared. Such comparison would warrant a paragraph in the discussion and could be potentially insightful.

We have included a comparison of TRMT10C NTD interaction with either ELAC2 or PRORP in Figure EV2B.

- Line 35: "tRNA punctuation model" not "punctate model"

Thank you for this observation. We have now changed this to "tRNA-punctuation model" (compound modifier).

- Line 37: Hällberg & Larsson, 2014 is an inappropriate self-reference here. A newer review comparing the different forms of RNase P would be more appropriate here (e.g., 10.1007/978-3-030-57246-4_11) or individual reviews of the two forms of RNase P (e.g., 10.1016/j.jbc.2024.105729 and 10.1016/j.jbc.2024.105731)

Although the differences between nuclear and mitochondrial RNase P:s are properly addressed in Hällberg & Larsson, 2014 we agree to replace it with a recent review that only focuses on the said difference and also includes structural biology aspects: Sridhara, 2024.

- Line 40: reference (Bartkiewicz et al., 1989) is in the wrong place; move behind the comma in line 39.

Thank you for this observation. We have now changed this.

- Line 42: add 10.1093/nar/gkad713, where this was finally demonstrated.

We have added this publication to the list of references here. As a note, PRORP is the nuclease that was presumed by Holzmann et al., 2008 based on sequence analysis,

and shown by extensive site-directed mutagenesis followed by activity assays by Reinhard et al., 2015.

- Line 44: reference missing; add 10.1093/nar/gks910 after "co-substrate". Thank you for this observation. We have now added this reference.

- Lines 46-47: This has recently been shown to be an artifact (see 10.1093/nar/gkad713).

Two independent publications (Reinhard et al., 2015) and (Li et al., 2015) on high-resolution X-ray structures of human PRORP found the active site to be disordered and not binding magnesium, although they were both crystallized in high concentrations of $MgCl_2$ (20 mM and 5 mM). The published work (Bhatta et al., 2021) on the cryo-EM structure of the human mitochondrial RNase P describes the process of active site ordering and conformational change of human PRORP upon complex formation in detail (section: Recruitment and activation of the nuclease subunit PRORP). Nevertheless, since this is not important for the structural work on ELAC2, which is the paper's focus, and the journal policy of keeping the introduction brief, we have removed this part.

- Line 50: "prokaryotes" is outdated terminology; better use bacteria and archaea. Thank you for this remark; we have now changed to bacteria and archaea.

- Line 52: better "in many eukaryotes" than "in most ...", as it is not yet clear how widespread the small form is in eukaryotes. Thank you for this remark; we have now modified the writing accordingly.

- Line 64: incomplete sentence!

We have now removed the incomplete sentence "Just like the other long-form RNase Z enzymes, ELAC2 consists of "

- Line 69: delete RNA from "nuclear RNA processing". Done

- Lines 72-73: ELAC1 is unlikely to compensate for ELAC2 deficiency in the nucleus, but there is another major nuclear tRNA 3'-end processing pathway that has to be mentioned in this context.

Mitochondrial dysfunction is the only validated effect of the clinically relevant mutations of ELAC2 that have been found so far. Therefore, the remaining activity of ELAC2 with the mutations and/or ELAC1 seems to be able to compensate (or other systems like the Rex1p pathway in yeast (not well characterized in human cells)) in the case of the

clinical mutations. It is essential to acknowledge that nuclear tRNA processing is not as clear-cut as one would think. See, for example, 10.1371/journal.pone.0154044.

Since our focus remains on the structural basis of mitochondrial 3'-tRNA processing, we have now removed the last piece of this sentence so that it reads:

“To the best of our knowledge, no clinically relevant mutations in ELAC2 have been reported related explicitly to nuclear RNA processing, presumably because mitochondrial dysfunction dominates the clinical picture.”

- Lines 130-133: I do not see how and why the similarity of the SDR5C1/TRMT10C/tRNA complex in the structures with PRORP and with ELAC2 would support or "highlight" a coupling of the 2 reactions. In fact, I do not see any evidence for a true coupling of the reactions.

The explanation is in the same sentence directly afterwards. “as it is only required to exchange the RNA processing catalytic subunit without major structural rearrangements in the SDR5C1/TRMT10C/tRNA platform.” Furthermore, in our hands, as well as others (Bhatta et al, 2024) SDR5C1/TRMT10C does not effectively release the tRNA after processing by PRORP.

- Lines 165-167: What is the Ψ -loop? I suppose it should either read T loop or T Ψ C loop.

We have now labeled this part of the tRNA consistently as a T loop.

Anyway, tRNA-His has a canonically sized T loop of 7 nt; only 2 mitochondrial tRNAs have T loop of 8 nt and one of 9 nt. I am not convinced that would lead to base stacking with F254 as suggested.

Since this is not studied structurally but merely a suggestive possibility, we have now removed the following text: “Furthermore, the exosite forms a hydrophobic pocket with F254 at the bottom. This pocket is vacant in our product ternary structure. Still, it might accommodate base stacking on F254 in mt-tRNAs with additional nucleotides in the T loop, and F254 is conserved in bilaterian ELAC2 homologs (Appendix Fig. S6A).”

- Lines 169-174: I am not convinced of the hydrophobic interactions, particularly in case of K260, because of the distances as apparent in the figure. One would require access to the PDB file to evaluate this.

We have now clarified this in the text by specifying “the alkyl chain of K260” (line 140). During the review period, we provided PDB files for examination by the reviewers through the editor. Rendering a Van-der-Waals surface shows no space for solvent between these hydrophobic residues (or parts of a residue when it comes to the alkyl chain of K260).

Are L103 and L104 in TRMT10C also conserved in metazoans? These two TRMT10C residues are conserved in mammals. We have included an MSA of TRMT10C at positions L103 and L104 in Appendix Fig. S5B and a reference to it in the results section.

- Lines 187-190: Are R788 and R791 in ELAC2, and Q343 in TRMT10C conserved in metazoans? ELAC2 R788 is conserved in metazoans, while R791 is conserved in mammals (Appendix Figure S5B). TRMT10C Q343 is conserved in metazoans. We have added the Appendix Figure S5D to show this.

- Line 193: Add reference (Rossmann et al., 1995) that first showed the processing order.
Done.

- Lines 252-258: To support the conclusions the acceptor stem of mt-tRNA-Ser(AGY) should be modeled; the cryo-EM densities are insufficient. We believe that the conclusions are well supported by the data but we have rephrased the text to highlight that the most likely explanation for the observed conformational change in the loop G156-P157 (which is included in the model) is the tRNA^{Ser}(AGY) contact.

- Lines 290-293 and Figure 5: The design of the experiment does not allow to distinguish between effects of trailer sequence CCA versus GAG (in the case of mt-tRNA-Ser(AGY)) and the different length of trailer. A tRNA-His with a GAG trailer should be used in comparison with the tRNA-His with CCA. That would also show whether the resolution of the gel as presented would at all have been sufficient to see cleavage of the 3 nt from the tRNA-His 3' end. This is a simple experiment that should be included. We have now replaced the gel in Figure 5A with the experiment suggested here.

- Lines 347-358: It has recently been shown that 5'-end processing of mt-tRNA-Ser(UCN) by PRORP is also stimulated by TRMT10C/SDR5C1 (10.1093/nar/gkad713). Thus, mt-tRNA-Ser(UCN) can obviously also be bound by TRMT10C/SDR5C1. Thank you, there is indeed a slight stimulatory effect by TRMT10C/SDR5C1 shown in supplementary figure 6B in 10.1093/nar/gkad713. Therefore, we have removed the "...,suggesting that it does not bind to TRMT10C/SDR5C1." part of that sentence.

Moreover, mt-tRNA-Gln requires TRMT10C/SDR5C1 for 5'-end processing by PRORP (10.1093/nar/gkad713), which is also at odds with the proposed model.

In this brief introduction to the structural work on the mitochondrial RNase Z, we have not proposed that mt-tRNA-Gln would not require TRMT10C/SDR5C1 in the mitochondrial RNase P reaction since there is no proof of that, and in the model proposal already seven years ago in Reinhard et al., 2017 there is no such suggestion either. To make this small passage crystal clear, we change:

“In this model, most tRNAs are recognized and bound by the TRMT10C/SDR5C1 complex, presumably co-transcriptionally, and then act as a maturation platform to stabilize the partially degenerate human mitochondrial tRNA pool during the ensuing maturation (Reinhard et al., 2017).

“

To:

“In this model, most, **if not all**, tRNAs are recognized and bound by the TRMT10C/SDR5C1 complex, presumably co-transcriptionally, and then act as a maturation platform to stabilize the partially degenerate human mitochondrial tRNA pool during the ensuing maturation (Reinhard et al., 2017).”

- Lines 360-364: As pointed out above, kinetic experiments would indeed be crucial to support the model the manuscript is built on. Speculations do not appear to be sufficient here.

We have now included an experiment in Figures EV1A and EV2A showing that TRMT10C/SDR5C1 stimulates processing by ELAC2. We think there are enough “likely” and “may” in these few sentences to flag that it is an informed speculation. Pre-organization of substrates is a poorly appreciated but important aspect of enzymology, especially when it comes to large and naturally quite flexible substrates like RNAs and proteins.

- Lines 390-400: Largely redundant with the results. Discussion could be shortened here.

We shortened this section by removing the following text: “Specifically, C1^{CCA} is recognized by H-bonding to Q727 and R728 in the sequence motif 724-HFSQRY-729 (Appendix Fig. S6E), where R728 is conserved, and Q727 is conserved in long-form RNase Z homologs.”

Dear Dr Hallberg,

Thank you for submitting a revised version of your manuscript. Your study has now been seen by all original referees, who find that their previous concerns have been addressed and now recommend publication of the manuscript. There remain only a few mainly editorial points that have to be addressed before I can extend formal acceptance of the manuscript:

- Please label the corr. author in ms file with email address provided
- Please rename the Conflict of Interest section into "Disclosure and Competing Interests Statement", in accordance with our updated Guide to Authors (<https://www.embopress.org/competing-interests>)
- Please adjust the in-text callouts for individual figures and figure panels: e.g. Fig. 2A-C appears to be missing.
- Please complete the general info table in the Autor checklist
- Please provide the source data for all EV and/or appendix figures, in a single ZIP
- Please provide the Synopsis image in jpg, png or tif format. The dimension should be exactly 550 pixels wide and between 300-600 pixels high.
- Please rename Table EV1 in ms file to Table 1 with the corresponding callout
- Please correct the section order. It should be: title page with complete author information, abstract, keywords, introduction, results, discussion, methods, data availability section, acknowledgements, disclosure and competing interests statement, references, main figure legends, tables, expanded figure legends.

With best regards,

Cornelius Schneider

Cornelius Schneider, PhD
Editor | The EMBO Journal
c.schneider@embojournal.org

We realize that it is difficult to revise to a specific deadline. In the interest of protecting the conceptual advance provided by the work, we recommend a revision within 3 months (8th Jan 2025). Please discuss the revision progress ahead of this time with the editor if you require more time to complete the revisions. Use the link below to submit your revision:

Referee #1:

In the revised version, the authors have thoroughly addressed all previous concerns and provided clear explanations on key points, particularly regarding ELAC2 efficiency, methyltransferase substrate identification, and the structural basis of tRNA processing order.

I am fully satisfied with the revisions and have no additional concerns. I recommend the manuscript for acceptance in its current form.

Referee #2:

The authors have addressed all my concerns and I think the quality of the manuscript has further improved.

The way the response is phrased made it hard to realize, where the authors performed additional experiments, introduced new figures and/or added new analyses. More precise descriptions in the response letter would have been appreciated. In addition, the absence of Figure numbers/legends and the movement of all figures to the end of the manuscript has made the reviewing of the work much more complicated than for the originally submitted version. I don't want to blame the authors for this, but I thought I mention it. I also was quite confused about the response to the second paragraph of my first issue -"We have now expanded the discussion to further clarify (lines 344-347)." ...until I realized that it should be lines 318-321.

Despite, the rather philosophical and not very satisfying response about the way forward with Frankenstein/composite cryo-EM maps, I am fine with supporting publication of the work in EMBO Journal.

I would also like to congratulate the authors.

Referee #3:

The manuscript has been revised and most issues addressed or commented in the response. I have no further comments.

- Please label the corr. author in ms file with email address provided

Done

- Please rename the Conflict of Interest section into "Disclosure and Competing Interests Statement", in accordance with our updated Guide to Authors (<https://www.embopress.org/competing-interests>)

Done

- Please adjust the in-text callouts for individual figures and figure panels: e.g. Fig. 2A-C appears to be missing.

Fig 2 was there but now modified to Fig. 2A-C. All instances of "Fig" were also changed to "Fig." according to the latest available style manual for EMBO Journal

- Please complete the general info table in the Autor checklist

Done

- Please provide the source data for all EV and/or appendix figures, in a single ZIP

Done

- Please provide the Synopsis image in jpg, png or tif format. The dimension should be exactly 550 pixels wide and between 300-600 pixels high.

Done

- Please rename Table EV1 in ms file to Table 1 with the corresponding callout

Done

- Please correct the section order. It should be: title page with complete author information, abstract, keywords, introduction, results, discussion, methods, data availability section, acknowledgements, disclosure and competing interests statement, references, main figure legends, tables, expanded figure legends.

Moved the abstract to start a new page to make the 1st page a title page and the expanded view figure legends to after Table 1.

Dear Dr. Hallberg,

I am pleased to inform you that your manuscript has been accepted for publication in the EMBO Journal.

Yours sincerely,

Cornelius Schneider

Cornelius Schneider, PhD
Editor
The EMBO Journal
c.schneider@embojournal.org
